

# The Horizontal Ice Nucleation Chamber HINC: INP measurements at Conditions Relevant for Mixed-Phase Clouds at the High Altitude Research Station Jungfraujoch

Larissa Lacher[1], Ulrike Lohmann[1], Yvonne Boose[1~], Assaf Zipori[2^], Erik Herrmann[3], Nicolas Bukowiecki[3], Martin Steinbacher[4] and Zamin A. Kanji[1]

[1] Institute for Atmospheric and Climate Science, ETHZ, Zurich, 8092, Switzerland

[2] Institute for Earth Science, Hebrew University, Jerusalem, 76100, Israel

[3] Laboratory of Atmospheric Chemistry, Paul Scherrer Institute, Villigen, 5232, Switzerland

[4] Empa, Swiss Federal Laboratories for Materials Science and Technology, Duebendorf, 8600, Switzerland

~now at Karlsruhe Institute of Technology (KIT), Institute of Meteorology and Climate Research, Garmisch-Partenkirchen, 82467, Germany

^now at Weizmann Institute of Science, Department of Earth and Planetary Sciences, Rehovot, 7610001, Israel

Correspondence to: Larissa Lacher (larissa.lacher@env.ethz.ch), Zamin. A. Kanji (zamin.kanji@env.ethz.ch)



**Abstract.** In this work we describe the Horizontal Ice Nucleation Chamber, HINC as a new instrument to measure ambient ice nucleating particle (INP) concentrations for conditions relevant to mixed-phase clouds. Laboratory verification and validation experiments confirm accuracy of the thermodynamic conditions of temperature (T) and relative humidity (RH) in HINC with uncertainties in temperature of ± 0.4 K and in RH with respect to water ($RH_w$) of ± 1.5 %, which translates to an uncertainty in RH with respect to ice ($RH_i$) of ± 3.0 % at T > 235 K. For further validation of HINC as a field instrument, two measurement campaigns were conducted in winters 2015 and 2016 at the High Altitude Research Station Jungfraujoch (JFJ; Switzerland, 3580 m a.s.l.) to sample ambient INPs. During winters 2015 and 2016 the site encountered free tropospheric conditions 92 % and 79 % of the time respectively. We measured INP concentrations at 242 K at water sub-saturated conditions ($RH_w$ = 94 %), relevant for the formation of ice clouds, and in the water supersaturated regime ($RH_w$ = 103-104 %) to represent ice formation occurring under mixed-phase cloud conditions. In winter 2015 and 2016 the median INP concentrations at $RH_w$ = 94 % was below the minimum detectable concentration. At $RH_w$ = 104 %, INP concentrations were an order of magnitude higher, with median concentrations in winter 2015 of 2.8 per standard liter ($stdL^{-1}$; normalized to standard temperature T = 273 K and pressure p = 1013 hPa) and 4.7 $stdL^{-1}$ in winter 2016. The measurements are in agreement with previous winter measurements obtained with the Portable Ice Nucleation Chamber, PINC, of 2.2 $stdL^{-1}$ at the same location. During winter 2015, two events caused the INP concentrations at $RH_w$ = 103-104 % to significantly increase above the campaign average. First, an increase to 72.1 $stdL^{-1}$ was measured during an event influenced by marine air, coming from the Northern Sea and the Norwegian Sea. Second, INP concentrations up to 146.2 $stdL^{-1}$ were observed during a Saharan dust event. To our knowledge this is the first time that a clear enrichment in ambient INP concentration is observed during a time of marine air mass influence, indicating the importance of marine particles on ice nucleation in the free troposphere.




## 1 Introduction

Clouds and aerosols continue to cause the largest uncertainty in the current assessment of global climate change (e.g. Boucher et al., 2013). Despite their importance in the Earth's system, fundamental knowledge on cloud formation and evolution is still missing. Clouds containing ice can have a positive or negative effect on the Earth's radiative budget, depending on their micro-
and macrophysical properties (Lohmann et al., 2016). Cloud microphysical processes are highly variable depending on the available amount of water vapor and the presence of supercooled cloud droplets and ice crystals. In addition, cloud microphysical processes can change during the development of a cloud, and the first formation of ice in clouds is still not completely understood.

Different processes leading to ice formation from the vapor or liquid phase are possible. In the absence of ice nucleating
particles (INPs) (Pruppacher and Klett, 1997; Lohmann et al., 2016) freezing of supercooled droplets occurs homogeneously, which is relevant in pristine atmospheric environments. It requires temperatures, $T < 235$ and a relative humidity (RH) with respect to ice ($RH_i$) > 140 % where the nucleation rate is high enough to outcompete heterogeneous nucleation. In the presence of INPs, heterogeneous ice nucleation is favored, since the particles can lower the energy barrier of the phase change. This freezing pathway is dominant in the mixed-phase cloud regime at temperatures $T > 235$ K, where ice and supercooled water
can co-exist and homogeneous freezing rates are negligible. Currently, four different heterogeneous freezing mechanisms are distinguished: Deposition nucleation, contact freezing, immersion freezing and condensation freezing (for a detailed description see Vali et al., 2015). Deposition nucleation is relevant for the formation of cirrus clouds as inferred by Cziczo et al. (2013), but is typically not relevant for the formation of mixed-phase clouds, as lidar observations show that the liquid phase is present before ice crystals form (Ansmann et al., 2008). The two most likely freezing modes in mixed-phase clouds
are immersion/condensation freezing, where the INP initiates the freezing from within a supercooled droplet. At present it is questioned if there is a physical difference between immersion and condensation freezing (Welti et al., 2014; Wex et al., 2014; Vali et al., 2015) but as the ice germ should form from the liquid phase in both cases, it is not expected so.

In addition to different possible ice formation pathways, the identification of ambient INPs remains challenging, since only a small fraction of aerosol particles (~1 out of $10^5$) nucleates ice (Rogers et al., 1998; DeMott et al., 2010), and the exact
properties rendering them ice-active are not known. INPs can be solid and water-insoluble, or also soluble and crystalline (Kanji et al., 2017), and their ice nucleation ability has been linked to a crystal lattice match to ice, surface defects which increase the density of adsorbed water molecules locally, or by functional groups which increase the chemical affinity to ice via hydrogen bonds (Pruppacher and Klett, 1997). From the large variety of ambient aerosol classes (Kanji et al., 2017), mineral dust particles were observed to nucleate ice below 258 K (Hoose and Moehler, 2012 and references therein), and it
has been found that K-feldspars are the most efficient INPs out of many tested minerals (Atkinson et al., 2013; Yakobi-Hancock et al., 2013; Zolles et al., 2015; Harrison et al., 2016; Kaufmann et al., 2016). Furthermore, due to its abundance in the lower free troposphere (FT), it is thought that mineral dust plays a key role in atmospheric ice nucleation (e.g. DeMott et al., 2003a; Kamphus et al., 2010). Particles of biological origin, like certain bacteria, fungal spores and pollen, were found to be efficient



INPs at temperatures above 263 K (Hoose and Moehler, 2012), however, the atmospheric concentration from whole and intact biological particles which are ice-active is temporally and spatially variable, and their influence on ice formation is therefore rather seasonal and local in nature (Després et al., 2012). Nanometer scaled fragments from biological particles are present in much higher concentrations and might have an atmospheric implication (Pummer et al., 2012; Augustin et al., 2013; O'Sullivan et al., 2015; Fröhlich-Nowoisky et al., 2015; Wilson et al., 2015). Additionally, bacteria have been found in Saharan (Meola et al., 2015) and soil dust aerosols (Conen et al., 2011), possibly influencing their ice nucleation activity. The role of marine aerosol as a source of INPs has been reported for the first time more than five decades ago (Brier and Kline, 1959; Cziczo and Froyd, 2014) and has been re-emphasized recently and observed in various field and laboratory studies (Cziczo et al., 2013; Knopf et al., 2014; Wilson et al., 2015, DeMott et al., 2016). Recent field studies or studies of field samples in the laboratory (Cziczo et al., 2013; Knopf et al., 2014; Wilson et al., 2015; Ladino et al., 2016; DeMott et al., 2016) have shown that particles and organic matter sampled or emitted from the sea surface can be a source of INPs. Marine aerosols are produced via a bubble-bursting mechanism (e.g. de Leeuw et al., 2011;Gantt and Meskhidze, 2013; Aller et al., 2005; Cunliffe et al., 2013) when entrained air bubbles rise through the sea surface microlayer and burst upon contact with the atmosphere. The sea surface microlayer is usually enriched in biogenic material leading to the emission of these in the atmosphere as aerosol particles. A source of these marine particles can be microorganisms like phytoplankton and bacteria, exopolymer secretion, colloidal aggregates, glassy organic aerosols, crystalline hydrated sodium chloride particles and frost flowers (summarized in Burrows et al., 2013). Cells or cell fragments and exudates of phytoplankton species were found to be ice-active (Knopf et al., 2011; Alpert et al., 2011;Wilson et al., 2015), and biological material during phytoplankton blooms might also play an important role for ice nucleation (Prather et al., 2013; DeMott et al., 2016). These marine aerosols can be sub-micrometer in size (e.g. 0.02 – 0.2 µm, Wilson et al., 2015; 0.25 – 1 µm, DeLeon-Rodriguez et al., 2013), a size range which is transported to higher altitudes. Burrows et al. (2013) state that marine biogenic particles are an important source of INPs in remote marine areas in the absence of other efficient INPs such as mineral dust. During airborne measurements, Cziczo et al. (2013) found sea salt in ice residuals from tropical tropopause cirrus clouds, especially over the open ocean, but also in reduced concentrations over land.

In addition to laboratory studies, which aim to understand the physical processes of ice nucleation and determine key aspects of aerosols acting as INPs, it is crucial to quantify the total number concentration of ambient INPs in an environment relevant for clouds containing ice and to address the question of their variability in space and time. Several studies exist from airborne platforms (e.g. Bigg, 1967; Rogers et al., 1998; Prenni et al., 2009; DeMott et al., 2010; Avramov et al., 2011, Schrod et al., 2017), and ground-based observations (e.g. DeMott et al., 2003b; Chou et al., 2011;Ardon-Dryer and Levin, 2014; Mason et al., 2016; Boose et al., 2016a; Boose et al., 2016b) quantifying the number concentration of INPs and their potential sources. Typically, filter sampling with subsequent offline freezing methods, and online measurements with continuous-flow-diffusion chambers (CFDCs) are used as INP measurement techniques. For filter sampling, aerosols are collected for a certain time and known air volume, after which the collected particulate is cooled and exposed to controlled temperature and RH conditions (e.g. Bigg, 1967; Santachiara et al., 2010; Conen et al., 2011; Bingemer et al., 2012; Ardon-Dryer and Levin, 2014; Knopf et





al., 2014; Mason et al., 2015). Filter techniques observe the onset freezing temperature of a sample with a very large number of particles resulting in a very sensitive detection limit. However, this comes at the cost of a low temporal resolution since the sampling times of the filters often are on the order of a few hours or longer. CFDCs measure INP concentrations in real-time with a higher temporal resolution, on the order of a few to tens of minutes (e.g. Rogers, 1988; Rogers et al., 2001; Chou et al.,

2011), but their total sampling volume is lower, and their sensitivity to detect INPs is limited at low concentrations (Boose et al., 2016a). This in particular is challenging at low supercooling or in areas where INP concentrations are lower than $0.1 - 1$ per standard liter (stdL$^{-1}$; normalized to standard temperature T = 273 K and pressure p = 1013 hPa).

Measurements of INP concentrations during different seasons in an environment which is relevant for the formation of mixed-phase clouds are rare. Conen et al. (2015) collected filters at different elevations in the (partly) FT, namely at Mt. Chaumont

and at the Jungfraujoch (JFJ) in Switzerland (1171 m and 3580 m, respectively) and at the Izaña observatory on Tenerife, Canary Islands (2373 m). They sampled for one year with one sample representing 24 hours, and INP measurements were reported for the temperature range 265 K to 269 K. At the JFJ, and for the same temperature range, they found a seasonal cycle with INP concentrations ranging from 0.001 to 0.01 stdL$^{-1}$, with a maximum in summer and a minimum in winter. This variation is attributed to previous INP activation and subsequent fall-out from advected air masses prior to reaching the JFJ,

leaving an air mass which is depleted in INPs upon arrival to the JFJ. Also at JFJ, INP measurements were performed at temperatures T = 241 K during winters 2012, 2013 and 2014 with the CFDC, Portable Ice Nucleation Chamber (PINC; Boose et al., 2016a). INP concentrations were sampled in the deposition nucleation mode in winters 2012 – 2014, and in winter 2014 also in the condensation freezing mode. Median INP concentrations below (above) water saturation were in the range of ≤0.05 – 0.1 (4.2) stdL$^{-1}$. To extend these measurements and to establish a longer time series of measurements at JFJ, INP

concentrations are measured since summer 2014 at the same temperature and RH conditions, with the newly built Horizontal Ice Nucleation Chamber (HINC) based on the design of Kanji and Abbatt (2009). In this study, the new chamber is characterized to be used not only for laboratory studies, but also as a field instrument. To complement the validation and verification experiments performed in the laboratory, two field campaigns in winters 2015 and 2016 were performed, and results are compared to previous winter measurements from the same location discussed in Boose et al. (2016a). In addition,

two events of anomalously high INP concentrations from the winter 2015 campaign are discussed to investigate the origin of these INPs.



## 2 Ice nucleation measurements

### 2.1 Technical description

HINC is a continuous flow thermal gradient diffusion chamber, based on the design of the UT-CFDC (Kanji and Abbatt, 2009). A schematic of HINC is shown in Fig. 1, including the outer dimensions of the chamber. Inner dimensions and more

detailed design aspects can be found in Kanji and Abbatt (2009). HINC consists of two horizontally oriented copper plates which are cooled by an external re-circulating ethanol cooler (LAUDA, RP 890 C). Self-adhering glass fiber filter papers (PALL 66217) mounted on the inner walls of the chamber are wetted prior to an experiment to create an ice layer upon cooling the walls. For the wetting procedure, the wall temperatures are kept at room temperature and the chamber is tilted to an angle of 45°, and approximately 100 ml of double-deionized water is used to wet the filter papers via four water ports, which are in

contact with the filter papers of the cold and warm wall. The chamber is kept in this position for 30 minutes to drain excess water via the outlet port downstream of the chamber. After draining, the chamber is brought back into a horizontal position and the outlet port is dried to ensure no residual water drops are retained. Following the wetting procedure, an optical particle counter (OPC, MetOne, GT-526S) is attached to the outlet port, and the walls are cooled down to the desired set point temperature T < 273 K. To establish a $RH_i$ and $RH_w$ > 100 %, a temperature gradient $\Delta T$ is applied between the two ice coated

walls (both walls below 273 K), with the upper wall set to the warmer temperature. The horizontal orientation of the chamber ensures no internal convection. For an experiment where the RH should be increased at a constant center temperature, which is typical to determine the onset RH of INPs, the $\Delta T$ is achieved by a temperature increase and decrease of the respective walls at equal rates. This ensures that the temperature in the center, where the air containing the aerosols is injected, remains constant. Aerosol enters the chamber via a movable injector, and the aerosol flow is layered in between a dry particle-free sheath nitrogen

(purity 5.0, 99.999 %, $H_2O \leq 3$ ppm) flow, with a sheath-to-aerosol flow ratio of typically between 10:1 and 12:1, which ensures the aerosol flow remains laminar and is exposed to the constant center temperature and RH conditions in the chamber. The sheath air is controlled by a mass flow controller (MFC; MKS, MF1, full scale flow of 5 stdL min$^{-1}$). The position of the injector thereby determines the residence time of the aerosols in the chamber. Within this residence time the aerosols can nucleate into ice crystals and grow to larger sizes, allowing for discrimination by size with the OPC. For the field measurements

reported here, particles detected by the OPC in size bin > 5 μm in diameter are classified as ice (see sect. 2.2.3). Counts in smaller size channels are contaminated by unactivated aerosol particles and, at water saturated conditions, the size bins up to 3-4 μm could be contaminated by droplets. For measurements reported here, HINC was kept at sufficiently low RH to ensure water droplets did not contaminate the signal in the 5 μm channel. The OPC is calibrated for a total flow of 2.8 stdL min$^{-1}$ which is set by an external pump. The MFC is used to set the 92 % sheath air flow, so that the remaining 8 % is made up by

the aerosol flow sampled (pulled) into the chamber.

LabVIEW® is used to control the re-circulating cooler temperature and resulting RH in the chamber by regulating the cold and warm wall temperatures. Integrated into the LabVIEW® control panel are the flow rate of the sheath flow through the





MFC and a motorised valve to direct aerosol the aerosol flow through a HEPA filter to quantify the noise for the signal to noise ratio (see sect. 2.3). Additionally, the counts in all size bins of the OPC are read out, and all set and output parameters are logged into a single file which is later used for data analysis.

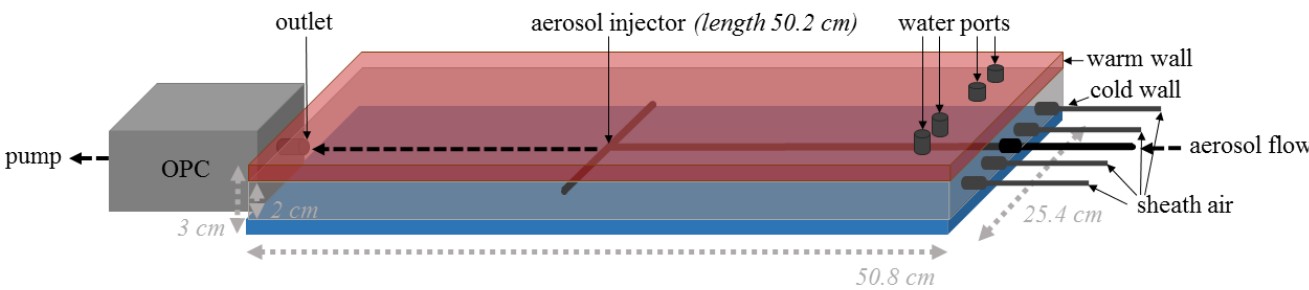

Fig. 1: Schematic and outer dimensions of HINC (main chamber).

## 2.2 HINC validation and verification

Here we present laboratory measurements, using a variety of aerosol particles, to verify the accuracy in temperature T and RH
for the newly built INP counter. We define the operational range, where it reliably measures INPs in the water saturated regime. To confirm these operation settings in the field, additional tests with ambient particles at the JFJ were performed.

### 2.2.1 Sample preparation

To validate the temperature and RH conditions in HINC, hygroscopic growth upon deliquescence, cloud droplet activation and
homogeneous freezing experiments with size selected ammonium sulfate ($(NH_4)_2SO_4$), sodium chloride (NaCl) and sulfuric acid ($H_2SO_4$) particles were conducted, as well as experiments with polydisperse ambient particles in the field. Except the latter, particles were generated as aqueous solutions (0.05625 % w/w for $(NH_4)_2SO_4$, 5 % w/w for NaCl, 60 % w/w for $H_2SO_4$), atomized and dried by diffusion to $RH_w < 2$ %, before they were size selected by a differential mobility analyzer (DMA; TSI, 3081). For the deliquescence and cloud droplet activation experiments 200 nm particles, and for homogeneous freezing
experiments 100 nm $H_2SO_4$ were used. The activated fraction (AF) which is the ratio of aerosol particles which activated into cloud droplets or nucleated to ice crystals to the number of total particles, counted by a condensation particle counter (CPC; TSI 3772) in parallel, are reported.





### 2.2.2 Accuracy of Temperature and RH in HINC

At temperatures T < 235 K, homogeneous freezing experiments were conducted to compare the onset of freezing observed in HINC to those reported and modeled in the literature (Koop et al., 2000a). Experiments with 100 nm $H_2SO_4$ particles at 233 K (Fig. 2) revealed that the onset of freezing occurs within $RH_w \pm 1.5$ % compared to the expected $RH_w$ of the freezing of solution droplets of the same initial dry size of the solute particles (Koop et al., 2000a) as shown by the dashed line in Fig. 2.

Above homogeneous freezing temperatures (T > 235 K), we expect cloud droplet activation for 200 nm $(NH_4)_2SO_4$ and $H_2SO_4$ particles at $RH_w = 100$ %. This is observed in the smaller OPC channels (0.5 – 2 μm) as an increase in the AF when the conditions in the chamber approach $RH_w$ 99-100 % as observed for $H_2SO_4$ (Fig. 3) and for ambient particles (Fig. 4). We note that an increase in the AF of $H_2SO_4$ is observed prior to $RH_w = 100$ % in the 0.5 and 1 μm channel (Fig. 3) which is to be expected due to hygroscopic growth of the $H_2SO_4$ particles. Therefore the increase at $RH_w < 100$% is only observed in the smaller size channels which should occur prior to droplet activation at $RH_w = 100$ %, while a clear increase at $RH_w = 100$ % in the 2μm channel is observed due to cloud droplet activation. On the other hand, the ambient particles show a delayed droplet activation in the 2μm channel at $RH_w = 101.5$ % (Fig. 4). This is likely due to the lower hygroscopicity of the ambient particles compared to $H_2SO_4$ but could also be compounded by RH uncertainties (see sect. 2.3).

At lower $RH_w$ hygroscopic growth due to deliquescence was also observed as an increase in particle concentrations in the smallest two OPC channel of 0.3 and 1 μm, which occurred for 200 nm NaCl in the range of $RH_w$ 79.5 – 81 % (Fig. 5). The observed increase in AF due to deliquescence and hygroscopic growth compares well to literature results reported to be $RH_w = 77 \pm 2.5$ % (Koop et al., 2000b). Due to the generation method of NaCl at T > 273 K and immediate exposure of the NaCl to the respective temperature and humidity, when the aerosol particles are injected in HINC, we believe that we deliquesce NaCl anhydrate, and not NaCl dihydrate (Bode et al., 2015). For 200 nm $(NH_4)_2SO_4$ particles, deliquescence and hygroscopic growth was also observed at $RH_w = 82-85$ %, consistent with the $(NH_4)_2SO_4$ deliquescence $RH_w = 82 – 84$ % (Cziczo and Abbatt, 1999). All validation experiments to verify the RH and temperature accuracy in HINC are summarized in Fig. 6.





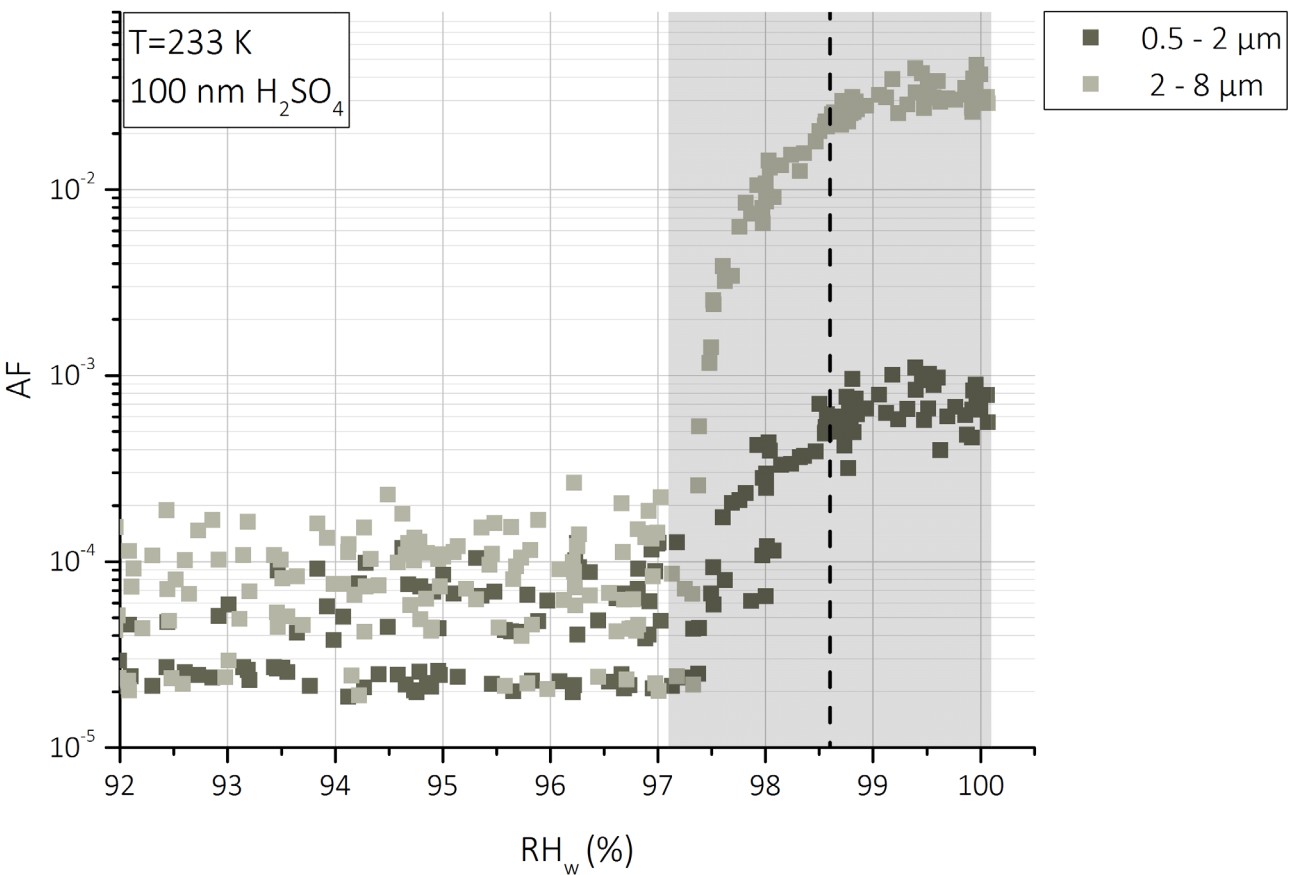

Fig. 2: Homogeneous freezing curve of 100 nm dry diameter $H_2SO_4$ particles, shown as the AF as a function of $RH_w$ at 233 K. The dashed line ($RH_w$ = 98.6 %) represents the expected $RH_w$ for homogeneous freezing of dilute solution drops of an initial dry diameter of 100 nm (Koop et al., 2000a). The shaded region indicates the range of uncertainty in $RH_w$ in HINC (see sect. 2.3)





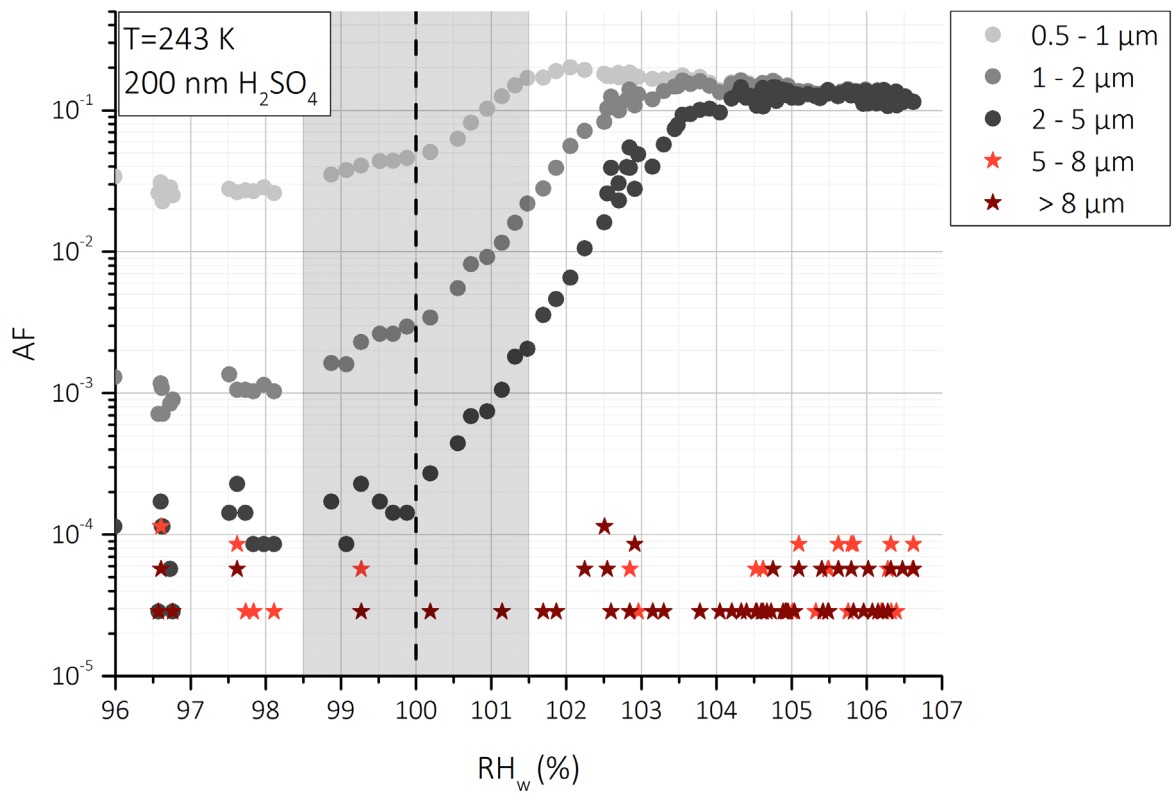

Fig. 3: Water droplet activation fraction and subsequent growth as function of $RH_w$ at 243 K for 200 nm dry diameter $H_2SO_4$ particles, shown for all size channels > 0.5 μm in the OPC.





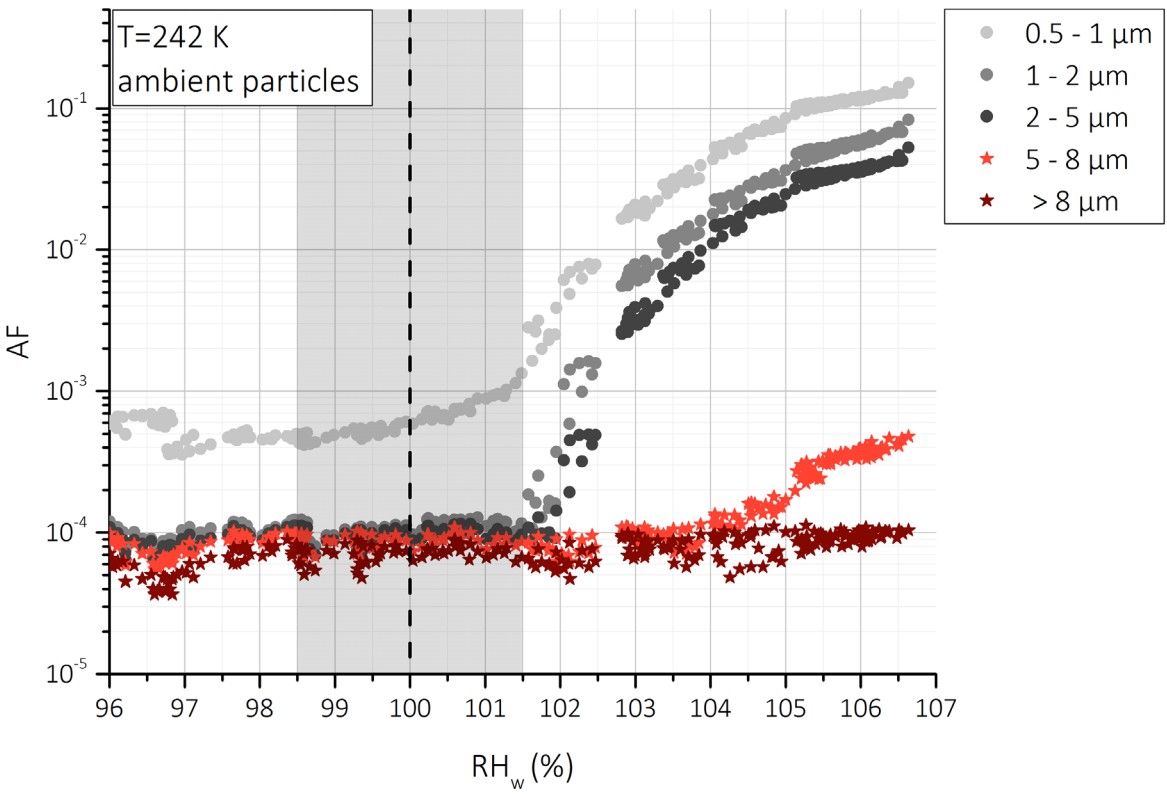

Fig. 4: Water droplet activation fraction and subsequent growth as function of $RH_w$ at 242 K for ambient polydisperse particles sampled at the JFJ, shown for all size channels in the OPC.





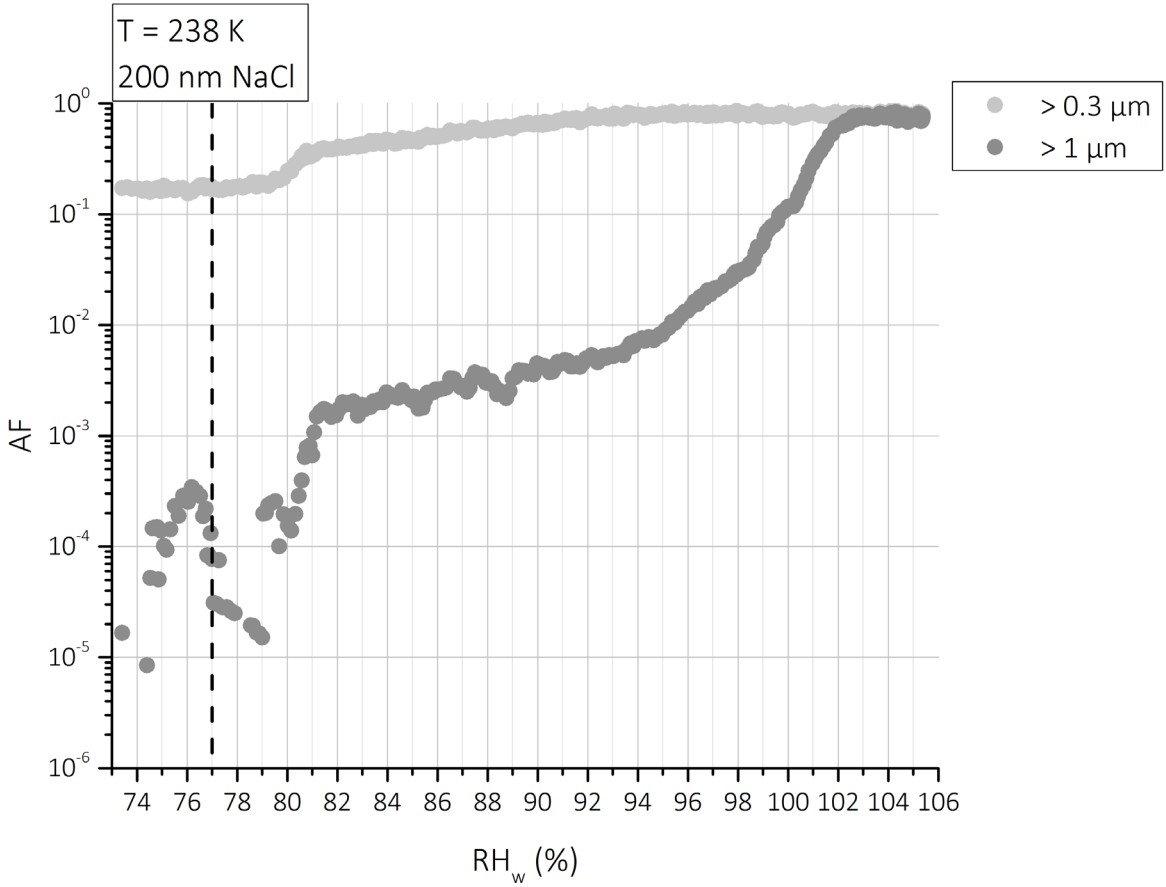

Fig. 5: AF showing deliquescence and subsequent hygroscopic growth as function of $RH_w$ at 238 K for 200 nm NaCl particles.

### 2.2.3 Upper RH limit for ice crystal detection: water drop survival (WDS)

5   The upper RH achievable to reliably detect ice crystals is limited by the possible activation and subsequent diffusional growth of water drops, since only the optical size is used to discriminate between ice crystals (larger) and water droplets (smaller) at the same temperature and RH conditions. To identify the maximum operation RH, experiments are conducted with 200 nm $(NH_4)_2SO_4$ and $H_2SO_4$ particles for T > 235 K where homogeneous freezing is ruled out. For these experiments the RH is increased until activated water droplets grow to a diameter which is detected in the OPC size channel used to detect ice crystals.

10   This is referred to as the WDS point. An example of an increase in $RH_w$ to > 106 % at 243 K is shown in Fig. 3, where WDS is not observed in the OPC channel > 5 µm. This is likely due to settling of the larger liquid droplets out of the aerosol flow, which grow to sizes too large to be sampled by the OPC due to the hygroscopic nature of $H_2SO_4$. Note that the AF even at low RH is non-zero, which is caused by either unactivated sample particles (particularly in the smallest size channels) or by internal





background counts (see section 2.3). The same experiment was performed in the field with polydisperse ambient particles at 242 K (Fig. 4). This test reveals an increase in AF of a factor of 4 in the OPC size channel 5 – 8 µm at $RH_w$ = 105 %, presumably caused by growing water droplets. However, in this test ice crystals are likely contaminating the signal and cannot be completely ruled out since the exact composition of the ambient particles is not known. This tests together with the $H_2SO_4$

tests, shown in Fig. 3, give us confidence that by operating at a temperature T = 242 K and a $RH_w$ ≤ 104 %, the OPC size channel > 5 µm is suited to reliably detect ice crystals in ambient conditions. The field measurements above water saturation presented in this work are therefore conducted at $RH_w$ 103 – 104 %.

### 2.2.4 Summary of validation and verification experiments

In Fig. 6, we show the results from all the validation experiments performed. The observed phase changes are in the expected range of homogeneous freezing of solution droplets (Koop et al., 2000a), cloud droplet formation (Lohmann et al., 2016), and deliquescence of NaCl (Koop et al., 2000b) and $(NH_4)_2SO_4$ (Cziczo and Abbatt, 1999). The ice onset for homogeneous freezing of $H_2SO_4$ was observed within a range $RH_w$ ± 1.5 % of the value reported in Koop et al. (2000a). At colder temperatures, the onset of homogeneous freezing is observed to shift to higher $RH_w$ but still remains well within the range of uncertainty in RH

(see sect. 2.3) of HINC. Above 235 K, where freezing of dilute water droplets is not expected, droplet formation for both $H_2SO_4$ and $(NH_4)_2SO_4$ as well as ambient particles was observed at $RH_w$ = 100 ± 1.5 % where particles initially not detectable in the OPC size channel > 1 µm activate to droplets and grow large enough to be detected. At lower $RH_w$ we observe hygroscopic growth upon deliquescence of NaCl and $(NH_4)_2SO_4$ as an increase in the OPC size channel > 0.3 µm. For NaCl such an increase was observed at RHw = 79 – 81 %, as compared to a $RH_w$ = 77 ± 2.5 % (Koop et al., 2000b). For $(NH_4)_2SO_4$

a growth was observed $RH_w$ = 81 – 85 % at as compared to $RH_w$ = 82 – 84 % (Cziczo and Abbatt, 1999).

Both the phase change and cloud droplet formation experiments with $H_2SO_4$, $(NH_4)_2SO_4$, NaCl and ambient particles verify that HINC operates reliably at the discussed settings of temperature and RH.




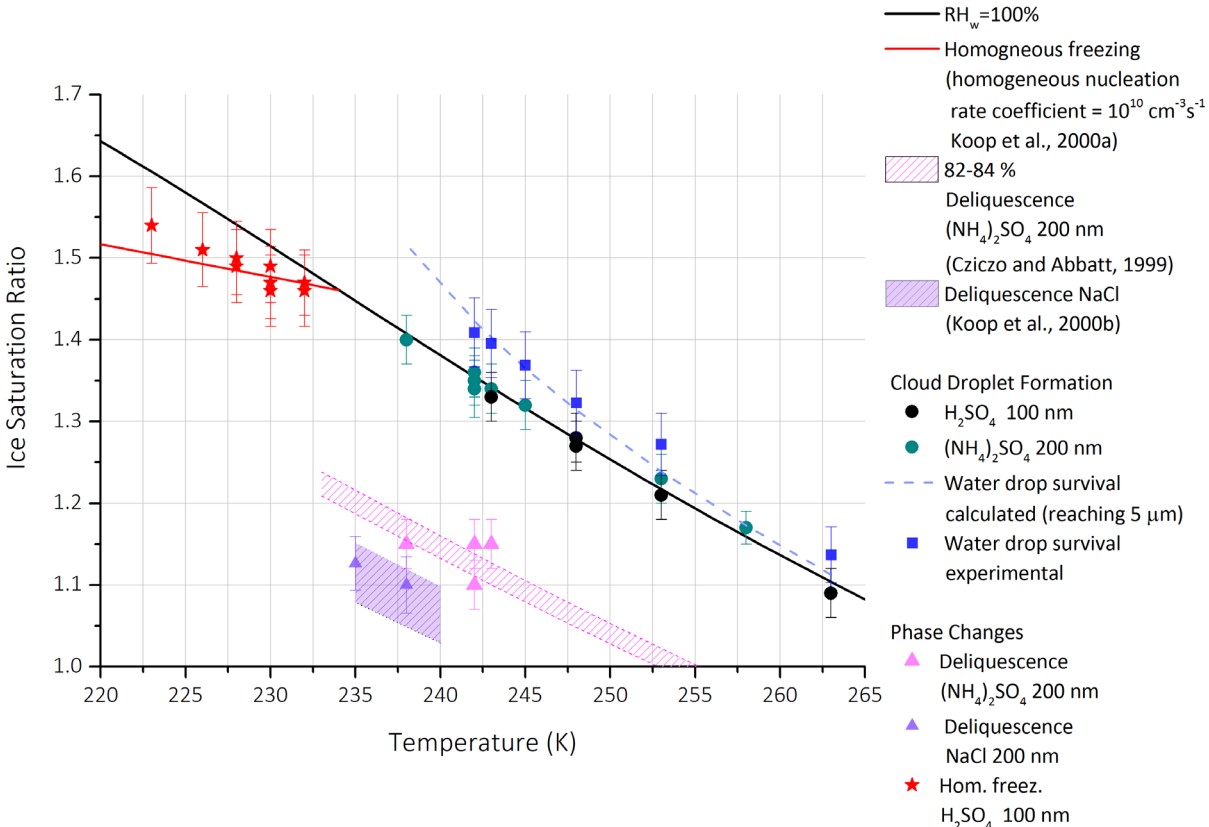

Fig. 6: Summary of characterization experiments and comparison to literature and theoretical values of phase changes and growth processes. Data from experiments reflect the first appearance of ice (ice onset) when the AF increases above the chamber background levels.

## 2.3 Uncertainties and limit of detection (LOD)

Uncertainties in temperature and RH that the aerosol particles are exposed to can arise from the set wall temperatures of HINC that are measured by two thermocouples on each wall in the activation/growth section of the chamber. The thermocouples have an uncertainty of $\pm 0.1$ K. This translates into an uncertainty in RH at the center location of $RH_w \pm 1$ % ($RH_i = \pm 2$ %) at

10 a center temperature T = 242 K and at $RH_w = 104$ %. In addition to this, the aerosol lamina is only $1/13^{th}$ of the total flow (sheath-to-aerosol ratio of 12:1), and since a linear temperature gradient establishes between the warm and the cold wall, there is a temperature variation of $\pm 0.4$ K across the aerosol lamina for the temperature conditions (242 K) used in the field measurements presented here. The variation in temperature relates to a variation of $RH_w \pm 1.2$ % ($RH_i \pm 2.7$ %). Considering the validation experiments of homogeneous freezing and cloud droplet activation also reveal an absolute upper limit





uncertainty, we take the temperature and $RH_w$ uncertainty to be $\pm 0.4$ K and $\pm 1.5$ % ($RH_i \pm 3$ %) respectively which is slightly higher than the calculated uncertainties.

The OPC which is used to classify and count detected hydrometeors downstream of HINC has a relative counting accuracy of $\pm 10$ %, and a relative uncertainty in the sizing channels of $\pm 10$ %. The CPC which was used to measure the total particle

concentration in parallel to the INP measurements has a relative counting accuracy of $\pm 10$ %, resulting in a relative uncertainty in the AF of 14 %. The DMA which was used to size select the $(NH_4)_2SO_4$ and $H_2SO_4$ particles has a relative sizing uncertainty of $3 - 3.5$ %.

During an ice nucleation experiment, erroneous counts in the OPC ice channel can arise from electrical noise in the OPC, or from internal ice sources such as frost falling off the top, warmer chamber wall giving rise to particle counts that are falsely

classified as ice. In order to assess and to correct for the contribution of such false counts, filter measurements are conducted regularly before and after each sampling period to determine a background count in the following way: The instrument is set to its target temperature and RH, and the aerosol flow is sampled through a particle filter placed upstream of the aerosol injector for 10 minutes before and after each aerosol-sampling period of 20 minutes. It is observed that the background counts do not change significantly over this time and follow a Poisson distribution. The mean background ($\mu$) and the standard deviation ($\sigma$)

are derived from the two-10 minute filter periods. The mean ambient INP counts ($\overline{INP}$) are calculated from the mean counts ($\omega$) during a 20-minute sampling by taking the background into consideration, resulting in $\overline{INP} = \omega - \mu$. The $\mu$ and $\sigma$ are then used to assess if the signal, i.e. $\overline{INP}$, is significantly different from the noise. Therefore, the instrument's LOD for ambient INP following Poisson statistics is calculated as LOD = $\sigma$.

Comparing $\overline{INP}$ to the LOD can result in one of three scenarios:

1) $\quad \overline{INP} > \sigma$

2) $\quad \overline{INP} < 0$

3) $\quad 0 < \overline{INP} < \sigma$

Scenario 1 is considered as a quantifiable INP measurement with significance because $\omega > \mu + \sigma$ and therefore above the LOD. In scenario 2, INP counts are considered to be non-quantifiable for the volume of air sampled during the 20 minute period. In

scenario 3, INP counts are below the LOD but above $\mu$ therefore quantifiable but not with significant confidence. INP concentrations below the instrument LOD (scenario 2 and 3) are included in calculations of field campaign averages. We believe that this is crucial since ambient INP concentrations are typically quite low in the FT, and scenarios 2 and 3 occurring frequently actually speaks to this low observed INP concentration. Complete ignorance of values from scenarios 2 and 3 would lead to an artificially positive bias in reporting INP concentrations (for detailed discussion see Boose et al., 2016a). The

concentrations below the LOD are taken into account as their measured value (scenario 3). In the case of scenario 2, instead of using a value of zero for calculating campaign averages, a minimum quantifiable concentration for a 20 minute period is





used. This is determined by taking the minimum count possible in the OPC ice channel (1 count) and normalizing to the volume of ambient air during a 20-minute sampling period. By doing so we acknowledge that the true concentration could be below this minimum value (shown in Table 2). In addition, by accounting for the minimum quantifiable concentration in the manner described above, we take into consideration the increase in sampled volume of ambient aerosol flow due to use of an aerosol

5  concentrator, applied in winters 2013 and 2014 (Boose et al., 2016a), which lowers the LOD by a certain concentration factor. For transparency we show average INP concentrations including and excluding the values below the LOD. Finally, ambient INP counts and LODs are converted to concentrations in stdL$^{-1}$.



## 3 Field Measurements

To further validate the chamber performance for field measurement, INP measurements were conducted at the JFJ during the winters of 2015 and 2016 with HINC, and are compared to earlier measurements conducted at the same location and sampling conditions with PINC (Boose et al., 2016a) during winters 2012, 2013 and 2014. The ice nucleation measurements were

performed in the deposition nucleation mode, and since winter 2014 also in the condensation freezing mode. Detailed measurement dates and sampling periods for the campaigns are given in Table 1.

Table 1: Field measurement period and respective total sampling time of INP measurements and cloud water samples. Measurements were performed with PINC (winter 2012 – 2014) and HINC (winter 2015 and 2016).

| | | | | | total sampling time (h) | | |
| | | | | | PINC/HINC | | |
| measurements | start | end | breaks | T (K) | $RH_w$ 93/94 % | $RH_w$ 103/104 % | cloud water |
|---|---|---|---|---|---|---|---|
| winter 2012* | 12.01. | 27.01. | - | 241 | 62.3 | - | |
| winter 2013* | 24.01. | 27.02. | - | 241 | 138.9 | - | |
| winter 2014* | 24.01. | 16.02. | - | 241 | 54.6 | 28.5 | 67.8 |
| winter 2015 | 24.01. | 09.02. | - | 242 | 16 | 26 | 146 |
| winter 2016 | 13.01. | 06.03. | 01.02.-26.02. | 242 | 17.1 | 99 | 42.5 |

* Boose et al. (2016a) measured with PINC

HINC was setup in the field as shown in the schematic in Fig. 7. HINC sampled from a total aerosol inlet, which is described in detail by Weingartner et al., 1999. Ambient interstitial and cloud-phase particles with diameters < 40 µm and at wind velocities < 20 m s$^{-1}$ were sampled through an inlet heated to 293 K to evaporate cloud droplets and ice crystals. To exclude

an additional humidity source from ambient air, the aerosol flow was passed through a diffusion dryer ($RH_w$ < 2 %) and was then split into HINC (0.22 stdL min$^{-1}$ aerosol flow, 2.83 stdL min$^{-1}$ total flow) and a CPC (TSI 3772, 1 stdL min$^{-1}$), counting the total particle concentration in parallel.





The measurement conditions were set such that the aerosol flow experienced a constant temperature T = 242 K and a $RH_w$ of 94 % ($RH_i$ of 127 %), relevant for heterogeneous nucleation of ice clouds, and at T = 242 K and $RH_w$ = 104 % ($RH_i$ = 140 %), relevant for the mixed-phase cloud regime. The injector position was set to an optimal residence time (8 sec) for the aerosol particles, which takes into account prevention of ice crystal losses due to gravitational settling in the chamber but yet allowing

for enough growth time to reach an optical diameter of ≥ 5 μm. Experiments with two OPCs, one in parallel and one downstream of the ice chamber, were performed in order to obtain the difference in concentrations due to aerosol particle losses. For this test HINC was set to its field campaign temperature T = 242 K and well below water saturation to prevent any activation of particles as droplets or ice crystals. The experiments revealed a particle loss of 26 % for 1 μm particles, and 44 % for 2 μm particles. As such the INP measurements reported here are representative for ambient particles below 2 μm. This

size range is characteristic for ambient aerosols at JFJ, since particle concentrations > 1 μm are naturally very low (e.g. Nyeki et al., 1998, Baltensperger et al., 1998). In addition, calculations with the Particle Loss Calculator (von der Weiden et al., 2009) revealed that 0.8 % of 1 μm particles, and 2.6 % of 2 μm particles should be lost in the inlet and tubing upstream of HINC which we consider to be negligible in light of the uncertainty in particle counting from the OPC and CPC, and losses through the chamber injector.

For the field measurements, after the icing procedure (see sect. 2.1) the chamber walls were set to their target temperature, such that a center aerosol temperature T = 242 K and a $RH_w$ = 94 % was achieved. While the chamber cooled down to these conditions, it was flooded with filtered dry air to prevent moist room air from contaminating the iced chamber walls. When the target temperature and RH was reached, a 10-minute filter measurement, to quantify the background counts, was conducted, followed by a 20-minute aerosol measurement, and another 10-minute filter measurement. Following this procedure, the RH

was further increased to 104 %, and the background-sample-background measurement cycle was repeated.

Due to the non-automated operation of HINC, sampling time was limited to a maximum of 14 hours, of which approximately 50 % were performed during nighttime (7 pm - 7 am).



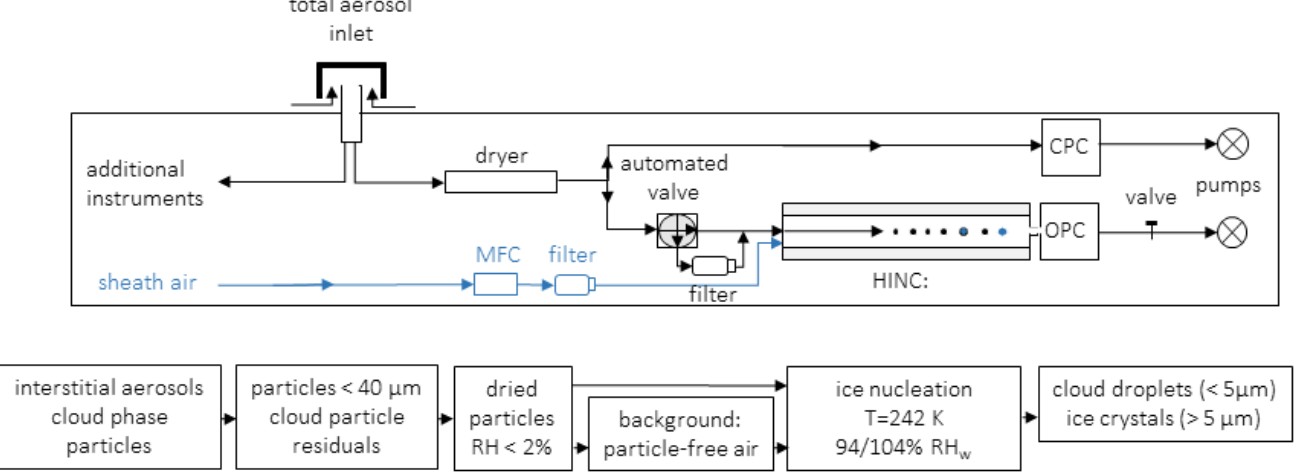

Fig. 7: Schematic of aerosol flow and instrument setup during field measurements at the JFJ research station.

## 3.1 Location

Measurements were performed at the JFJ, located in the Bernese Alps (3580 m a.s.l.; 46°33'N, 7°59'E). The research facility is a Global Atmospheric Watch (GAW) monitoring station and part of the ACTRIS2 Infrastructure (European Research Infrastructure for the observation of Aerosol, Clouds, and Trace gases), the Swiss National Air Pollution Monitoring Network (NABEL) and the SwissMetNet meteorological network. The station is located on an exposed mountain col, only surrounded by firn ice and rocks, without noticeable influence from local vegetation. Due to its elevation, the site is mostly located the FT and represents background aerosol concentrations (Baltensperger et al., 1997). It can be influenced by local emissions due to daytime tourist activities and boundary layer injections in the warmer season (Lugauer et al., 1998; Zellweger et al., 2003; Collaud Coen et al., 2011; Griffiths et al., 2013; Herrmann et al., 2015). In addition, the station is regularly affected by Saharan dust events (SDEs) where Saharan dust is transported within the FT to JFJ (Collaud Coen et al., 2004). Continuous measurements of aerosol physical properties (e.g. Baltensperger et al., 1997; Bukowiecki et al., 2016), trace gases (Steinbacher et al., 2017) and meteorological conditions (Appenzeller et al., 2008) are conducted and give additional information on aerosol properties and air mass origin to complement the INP measurements conducted.

## 3.2 Aerosol particle measurements

A custom built scanning mobility particle sizer (SMPS), consisting of a DMA (TSI 3071) and a CPC (TSI, 3775), measured the aerosol size distributions between 20 to 600 nm in diameter with a time resolution of 6 min (Herrmann et al., 2015). Larger





sized particles were measured by an OPC (GRIMM Dust Monitor 1.108; size range 0.23 – 16.4 μm). To merge the respective size distributions, the mobility and optical diameters were converted to volume equivalent diameters, assuming a particle density of 1565 kg m$^{-3}$ (Sjogren et al., 2008) and a unity shape factor. An integrating nephelometer (TSI, 3563) and an aethalometer (MAGEE scientific, AE31) measured the total aerosol scattering coefficients (three wavelengths) and the

absorption coefficients (seven wavelengths), respectively. From this the single scattering albedo (SSA) at 450 nm, 550 nm and 700 nm is derived, as well as the SSA Ångström exponent, as described by Collaud Coen et al. (2004). For the normal background aerosol, the SSA increases with wavelength, resulting in a positive SSA Ångström exponent, while for Saharan dust particles, due to their larger size and different optical properties, the SSA decreases with wavelength and its exponent becomes negative. A SDE is declared if the SSA exponent is negative for more than 4 consecutive hours. The aethalometer

also measures the equivalent black carbon (eBC) mass concentration, derived from the attenuation measurement by applying the factory standard mass attenuation cross sect. of 16.6 m$^2$ g$^{-1}$ at 880 nm.

### 3.3 Cloud water samples

Cloud water samples at JFJ were collected to determine the air mass origin and aerosol source regions by analyzing the samples

for trace chemical elements (e.g. Zipori et al., 2015), and as such were taken in parallel to the ice nucleation measurements during cloudy periods (see Table 1 for sampling times). Samples were collected on the terrace next to the laboratories at JFJ, with a home-built plexiglass plate (20x20x0.5 cm) which was attached vertically to the railing, facing the windward side. Prior to the sampling, the plate was cleaned with a super pure nitric acid solution (0.1 % w/w) and double-deionized water, and a blank sample with double-deionized water was taken by pouring it over the sampler. The sampling method works only for

supercooled cloud droplets which freeze upon contact with the plate, but not for ice crystals and precipitation particles, since ice crystals are deflected, and snowflakes are too heavy to stick to the sampler. The supercooled cloud droplets remain as an ice layer on the sampler, and is melted into pre-rinsed plastic bags and stored in Falcon® tubes. The samples were sent to the clean laboratory at the Hebrew University of Jerusalem, where they were analyzed for 23 trace metals concentration using inductively coupled plasma mass spectrometer (ICP-MS, Agilent, 7500cx). Detailed information regarding sample handling,

analysis protocol and quality control can be found in Zipori et al., 2012 and Zipori et al. (2015).

The interpretation of the analysis of elemental concentrations was focused on sodium (Na), aluminum (Al), lead (Pb) and strontium (Sr). Sodium chloride (NaCl) is a natural component of the ocean, and the positive ion Na$^+$ is found in an appreciable quantity in sea spray aerosol. In addition, Na$^+$ found in cloud samples is always accompanied by Sr. The fraction of Sr coming from sea salt (f(Sr)$_{ss}$) is used as an indication for air masses influence, and is calculated using the flowing equation:

$f(Sr)_{ss} = \left[\frac{Sr}{Na}\right]_{ss} \times \left[\frac{Na}{Sr}\right]_{samp}$  (1)



where, $f(Sr)_{ss}$ is the fraction of Sr contributed from sea salt, $[Sr/Na]_{ss}$ is the Sr to Na ratio found in sea salt, and $[Sr/Na]_{samp}$ are the concentrations of Na and Sr found in the samples (Herut et al., 1993). Furthermore, elemental ratios such Na/Al, Pb/Al and Pb/Na were used as indications for marine/dust, anthropogenic/dust and anthropogenic/marine influence in the samples, respectively.

In addition to the chemical analysis, Sr isotopic ratios were also measured with a multi-collector inductively coupled plasma mass spectrometer (MC-ICP-MS, NEPTUNE Plus). Sr separation was done with Sr-Spec resin flowing the method described by Stein et al., 1997. Since marine $^{87}Sr/^{86}Sr$ is constant with a value of 0.70917 (Hodell et al., 1990), while basalt and volcanic rocks has lower ratio and Saharan dust have higher ratio (Capo et al., 1998 and references therein), this parameter can be used to determine the prevailing aerosol type in the sample due to scavenging.

### 3.4 Back trajectories and source emission sensitivities

To obtain information on the trajectories of the air masses arriving at JFJ, an ensemble of 10-day back trajectories were calculated every 6 hours with the LAGRANTO model (Wernli and Davies, 1997), based on ECMWF Integrated Forecast System wind fields. Back trajectories at five different locations, one ending at JFJ and four displaced by 0.5° to the North and

South, are started at four different altitude levels of 654 hPa, 704 hPa, 604 hPa and 754 hPa. In addition, source sensitivities are derived from the Lagrangian particle dispersion model, FLEXPART, products browser at EMPA (http://lagrange.empa.ch/FLEXPART\_browser/; Stohl et al., 2005; Sturm et al., 2013; Pandey Deolal et al., 2014), which indicates the possible origin of the particles as a probability of the geographical regions from which the aerosol particles were emitted. It simulates the release of 50,000 particles every 3 hours at JFJ, and traces the particles backwards driven by ECMWF

Integrated Forecast System wind fields.

### 3.5 Assessment of free tropospheric conditions

Different proxies are used in this study to qualitatively assess the exposure of the site to the FT. The ratio of total reactive nitrogen ($NO_y$, as the sum of nitrogen oxide, nitrogen dioxide and its atmospheric oxidation products) to carbon dioxide (CO)

is commonly used as an indicator for boundary layer injections into the FT at elevated stations (Zellweger et al., 2003; Zanis et al., 2007; Pandey Deolal et al., 2013 ;Griffiths et al., 2014; Herrmann et al., 2015; Boose et al., 2016a). Both tracers are subject to emissions from anthropogenic sources, however, the $NO_y/CO$ ratio is decreases with increasing transport (aging) of the air mass as CO is inert within the timescale of interest (days) while the concentration decay rate of $NO_y$ is higher. Thus, a $NO_y/CO$ ratio of 0.0057 ppb/ppb was chosen to distinguish between FT conditions and boundary layer influence, in accordance

to the value reported for winter time measurements at JFJ by Zellweger et al. (2003). $NO_y/CO$ ratios below 0.0057 indicate FT conditions while an influence of boundary layer is likely for ratios above this value.





The concentration of particles > 90 nm was also used to identify FT conditions, since particles of this size are not formed in the FT, but are transported from the boundary layer, and therefore gives information on boundary layer influence (Herrmann et al., 2015). A threshold of 100 cm$^{-3}$ was chosen, below which the air mass is assumed to be free tropospheric. It should be mentioned that the concentration of particles > 90 nm can be influenced by the occurrence of larger sized dust particles, and

5 should therefore be considered with care during SDEs which are transported in the free troposphere.



## 4 Results

This is the first study where a chamber of HINC's design has been characterized and used for field measurements at conditions relevant to the mixed-phase cloud regime (T > 235 K and $RH_w$ > 100 %). An identical chamber (Kanji and Abbatt, 2009) has been used for online field studies (Ladino et al., 2016) and processing of re-suspended field samples (Wilson et al., 2015), at

233 K and water sub-saturated conditions. Here we compare results from two field campaigns in winter 2015 and 2016 to previously conducted INP measurements in the same season with PINC (Boose et al., 2016a). These results extend the time series of INP measurements below water saturation since winter 2012, and above water saturation since winter 2014 at JFJ, which also contributes to the monitoring of INPs during winter months.

In winters 2015 and 2016, air masses containing high INP concentrations were sampled, which were excluded from the

comparison of the campaign average INP concentrations. Furthermore, two such air masses from winter 2015 are discussed to relate the observed increase in INP concentrations to aerosol properties and air mass origin, and therefore examining the possible sources of ambient INPs. In winter 2015, the JFJ experienced FT conditions during 79 % of the sampling time, with no specific increase in INP concentrations during boundary layer influence, while in winter 2016 the site was 92 % of the time in the FT, and an event of increased INP concentration was observed during boundary layer influence, which is also excluded

from the comparison of campaign averages in this study.

### 4.1 Field measurements of INPs: winter 2015 and 2016

### 4.1.1 INP concentrations at water sub-saturated conditions (deposition nucleation)

Measurements below water saturation performed with HINC during winter 2015 and 2016 are shown in Fig. 8, and are

compared to the measurements performed with PINC in winters 2012 – 2014 taken from Boose et al. (2016a). Campaign median and mean INP concentrations are given in Table 2, panel 1 and 3. The solid boxes in Fig. 8 include the entire distribution of INP concentrations measured including those below the LOD, whereas the dashed ones include only the INP concentrations above the LOD (see sect. 2.3). Excluding INP below the LOD artificially positively biases the data to higher INP values, therefore in discussing the results below the averages that include INP concentrations below the LOD are considered. During

winters 2013 and 2014 an aerosol concentrator used which increased the signal-to-noise ratio by a factor of 3, and therefore the LOD for ambient INP was lowered.

The median (mean) INP concentration below water saturation, for all five field campaigns, is 0.1 (0.6) stdL$^{-1}$. The PINC measurements in winter 2012 – 2014 give a median (mean) INP concentration of 0.1 (0.2) stdL$^{-1}$, as compared to a HINC median (mean) concentration of $\leq$ 0.2 (1.2) stdL$^{-1}$ during winters 2015 and 2016. The natural variation in reported INP

concentrations at a given temperature and RH condition can be as much as an order of magnitude after accounting for





contributions from known INP sources The factor of 3-5 observed here between HINC and PINC suggests good agreement between the chambers given that the measurements are done in different years.

5   Fig. 8: Averaged INP concentrations observed below water saturation (see legend for exact temperature and RH conditions). Dashed boxplots represent only INPs > LOD, solid boxplots include INPs < LOD (see sect. 2.3 for details). Median: middle bar, mean: open square data point, box: interquartile range ($25^{th}$ to $75^{th}$ percentile), whiskers: $5^{th}$ and $95^{th}$ percentile. Data used to produce the distributions exclude contributions from measurements during periods of known air masses arrived at the JFJ (see sect. 4.2) with anomalously high INP concentrations. See Table 1 for field campaign sampling times.



Table 2: Campaign INP concentrations (stdL$^{-1}$) excluding known cases of high INP concentrations, as measured with PINC (winter 2012 – 2014) and HINC (winter 2015 and 2016). Values are given for each field campaign (rows 1 and 2) and as averages over the PINC and HINC campaigns (rows 3 and 4). Measurements above water saturation were not conducted prior to 2014. INP concentrations here consider data below the LOD, and therefore can differ from values reported in Boose et al. (2016a).

| RH$_w$ | INP (stdL$^{-1}$) | winter 2012 | winter 2013 | winter 2014 | winter 2015 | winter 2016 |
|---|---|---|---|---|---|---|
| 93/94% | median | < 0.05* | 0.1 | 0.1 | ≤ 0.2* | ≤ 0.2* |
| | mean | 0.2 | 0.2 | 0.3 | 1.7 | 0.7 |
| 103/104% | median | - | | 2.2 | 2.8 | 4.7 |
| | mean | - | | 4.2 | 5.0 | 8.2 |
| 93/94% | median | 0.1 | | | ≤ 0.2* | |
| | mean | 0.2 | | | 1.2 | |
| 103/104% | median | - | | 2.2 | 3.8 | |
| | mean | - | | 4.2 | 6.6 | |

**\*Averaged INP concentrations reported as the minimum quantifiable INP concentration, for data points that fall under scenario 2 (see sect. 2.3) which are included in the averaging as this minimum.**

### 4.1.2 INP concentrations at water saturated conditions (condensation freezing)

Figure 9 presents INP measurements above water saturation for winter 2014 (Boose et al., 2016a) and for winters 2015 and 2016 from this study. INP concentrations are typically higher by approximately a factor of 10 as compared to water sub-saturated conditions (Fig. 8) yielding a much higher signal-to-noise ratio with only few data points falling below the LOD. As such the differences between the solid and dashed box plots in Fig. 9 are small.

The median (mean) INP concentration in winter 2014 as measured with PINC was 2.2 (4.2) stdL$^{-1}$, and in winters 2015 and 2016, as measured with HINC, was 2.8 (5.0) and 4.7 (8.2) stdL$^{-1}$, respectively (Table 2, panel 2 and 4). In winter 2016, the INP concentrations are higher compared to the previous two winters. We explain this by the higher frequency of dust aerosol during that particular season. During the time of the measurements, two Saharan dust events were detected based on the SSA Ångström exponent criteria, but it is possible that the station was under the influence of dust particles without identification of a dust event which requires the condition of 4 continuous hours of a negative exponent of the SSA. This would imply that the majority of ambient particles were non-dust particles but dust could still contribute to the total aerosol loading. This is supported by source emission sensitivities derived by FLEXPART, indicating the Sahara as a source region for several days in addition to the declared SDEs, and by the concurrent increase in the particle concentrations > 0.4 μm, which is typical for dust. Therefore, this difference in INP concentrations between 2015 and 2016 can be explained by a natural inter-annual



variation. Despite this, the difference is only a factor of 2, which is considered low given the possible range of observed INP concentrations for a given temperature can be much higher e.g. Schrod et al. (2017).

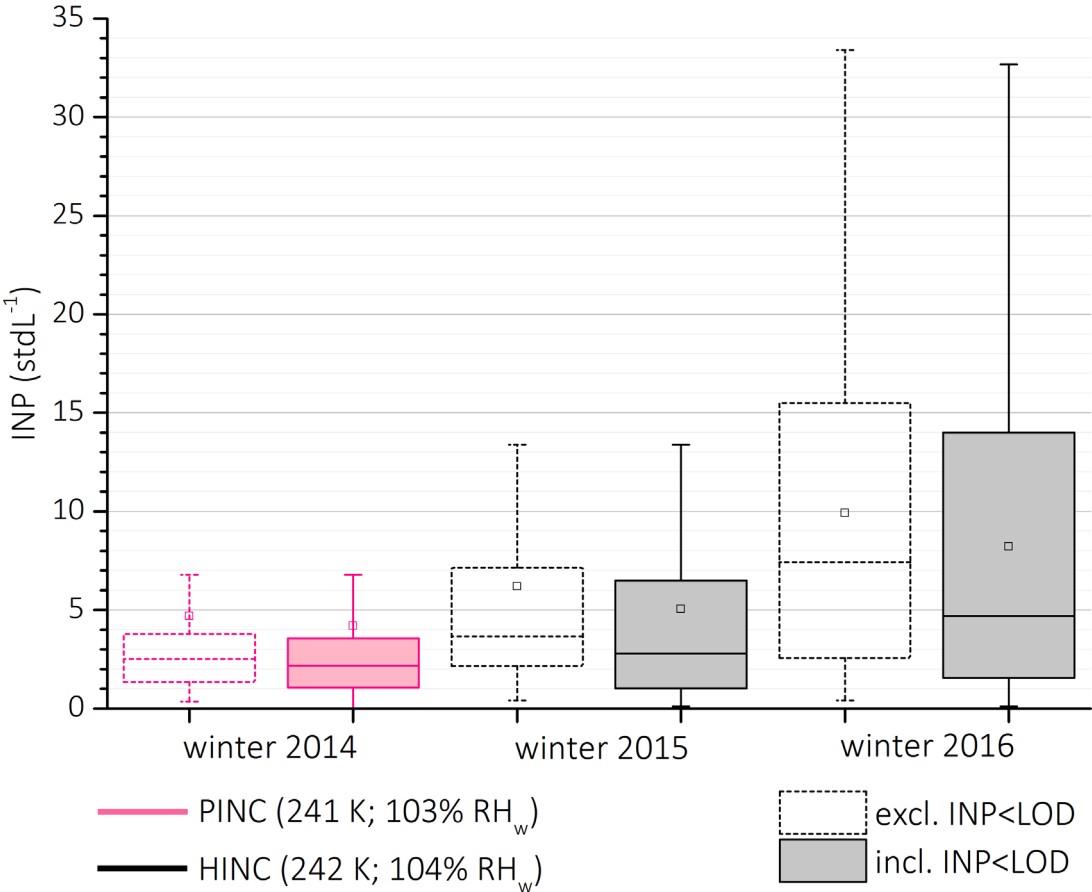

Fig. 9: Averaged INP concentrations observed above water saturation (see legend for exact temperature and RH conditions). Dashed boxplots represent only INPs > LOD, solid boxplots include INPs < LOD (see sect. 2.3 for details). Median: middle bar, mean: open square data point, box: interquartile range (25th to 75th percentile), whiskers: 5th and 95th percentile. Data used to produce the distributions exclude contributions from measurements during periods of known air masses arrived at the JFJ (see sect. 4.2) with anomalously high INP concentrations. See Table 1 for field campaign sampling times.



## 4.2 Case studies

INP measurements were performed with HINC from 24th January to 9th February 2015 at JFJ (Fig. 10). In this section, results from two events during winter 2015 are presented, for which INP concentrations above water saturation increased significantly above the campaign average The events which lasted several hours with higher INP concentrations observed is shown in Fig.

10 (panel a). We discuss the air mass characteristics and aerosol properties that identify the most likely sources of the observed increase in ice-active particles.

### 4.2.1 Case 2nd February 2015: marine air

On 2nd February INP concentrations increased up to 72.1 $stdL^{-1}$ with median (mean) concentrations of 16.4 (33.3) $stdL^{-1}$ over a time of 5 hours (Fig. 10, panel a). Also an increase in the AF (considering particles > 0.1 µm) by an order of magnitude was

observed from the campaign median (mean) value of  $1.3x10^{-4}$ ($2.8x10^{-4}$) to a maximum of $1.8x10^{-3}$ indicating that the increase was not only due to a new increase in particle number, but a higher fraction of the aerosol being ice active (Fig. 10, panel b).

The SSA showed a wavelength dependent increase, which is typical for background aerosol conditions at JFJ, and is an indication for the absence of Saharan dust. The particle concentration in the size bin 0.4 – 0.8 µm and > 0.8 µm did not increase (Fig. 10, panel c) confirming no influence from larger (dust) particles. The ratio Na/Al derived from the cloud water samples

during this sampling period increased (Fig. 10, panel d), and at the same time, $f(Sr)_{ss}$ increased (Fig. 10, panel d), which suggests that the air mass arriving at JFJ was of marine origin. Sr isotopic ratios, which could strengthen the identification of a marine source, were not available for that day, due to the small volume of cloud water collected which was not sufficient for the isotopic analyses. However, source sensitivities (Fig. 11, panel a) indicate most sources over the Northern Sea followed by the Northern Atlantic and Norwegian Sea. In addition to the source sensitivities, 10-day back trajectories also support marine

sources and also reveal that the air parcel travelled over Northern Europe, England and France to the JFJ (Fig. 11, panel c), and could have been subject to aging and anthropogenic emissions. Indeed on that day an increase in particle concentrations < 0.1 µm was observed (Fig. 10, panel c) which could be an indication for anthropogenic influence on the aerosol. This is supported by the finding of an increased $NO_y/CO$ ratio (Fig. 10, panel g), which is a tracer for anthropogenic influence of the air mass arriving at JFJ. Because eBC mass concentrations were low during that time (Fig. 10, panel g), a higher contribution

of marine particles than of the anthropogenic emissions to the observed INPs is likely. However, the influence of aging processes resulting in internal mixing of the marine particles with anthropogenic emissions arriving at JFJ cannot be ruled out.

The cloud sample analysis of the Na/Al ratio and the $f(Sr)_{ss}$, as well as the isotopic ratio of Sr (Fig. 10, panel d, e, f, respectively) revealed a marine source of particles on the morning of Jan 27th. Unfortunately, no INP measurements are available for that time, only later the same day, when the marine influence decreased, and INP measurements were within the campaign average.

Another marine air mass event was detected in winter 2016, on 6th March (data not shown here), when the indicators discussed above for marine influence (i.e. from cloud water samples) were similar. Using the same methods discussed above, we identified the INP concentration during the winter 2016 marine event to be in the same order of magnitude as during the winter





2015 marine event, with a median (mean) INP concentration of 14.9 (25.5) stdL$^{-1}$, and a maximum concentration of 176.8 stdL$^{-1}$.

In Fig. 12 we compare the INP concentrations from the periods of marine influence at JFJ to those reported in DeMott et al. (2016) for marine aerosols from online and offline INP measurements from different laboratory and field samples of sea waters.

5   We find good agreement at 242 K for the measured concentrations at JFJ during the two marine events, compared to the laboratory and field measurements of marine INPs. This is somewhat surprising given the measurements from DeMott et al. (2016) are taken directly over ocean sources, or from ocean waters in an ocean tank simulator. However, we note that during the occurrence of a marine event at the JFJ an increase in the AF (Fig. 10, panel b) is observed simultaneously, indicating an enrichment of ice active particles compared to the background levels of INP. Furthermore these observations might also

10  suggest that long range transport of small marine particles to the JFJ does not result in a suppression of their ice nucleation abilities, but rather the INPs retain their ice nucleation abilities during transport to the JFJ despite possible mixing with particles from anthropogenic emissions. Modelling studies have reported that marine aerosols as INPs are relevant on a global scale (Yun and Penner, 2013), especially in remote marine areas where dust abundance is low (Burrows et al., 2013; Wilson et al., 2015).





Fig. 10: Time series of (a) INP concentrations at 242 K and $RH_w$ 104 % and 94 % measured with HINC; (b) AF of $INP_{104\%}$ considering particles > 0.1 μm; (c) particle concentrations < 0.1 μm, 0.1 − 0.2 μm, 0.2 − 0.5 μm (SMPS) and 0.4 − 0.8 μm and >0.8 μm (OPC), given in volume equivalent diameter; cloud water sample analysis, given in ratios of mass of (d) Na/Al and $f(Sr)_{ss}$ and (e) Pb/Al and Pb/Na; (f) isotopic ratio $^{87}Sr/^{86}Sr$, the dashed line represents the constant value for marine sea salt (=0.70917); (g) eBC mass concentration and $NO_y/CO$ ratio.



Fig. 11: (a and b) FLEXPART emission sensitivity fields for 2nd (a) and 6th (b) February 2015, calculated 100 m above model ground level (http://lagrange.empa.ch/FLEXPART\_browser/; Stohl et al., 2005; Sturm et al., 2013; Pandey Deolal et al., 2014); the colour code represents the strength of source region contributions to the aerosol burden given in a unit flux per area.

5  (c and d) 10-day back trajectories for 2nd (c) and 6th (c) February 2015 calculated with LAGRANTO (Wernli and Davies, 1997); the colour code represents the trajectory pressure above model ground, black points indicate each 24-hour back calculation.





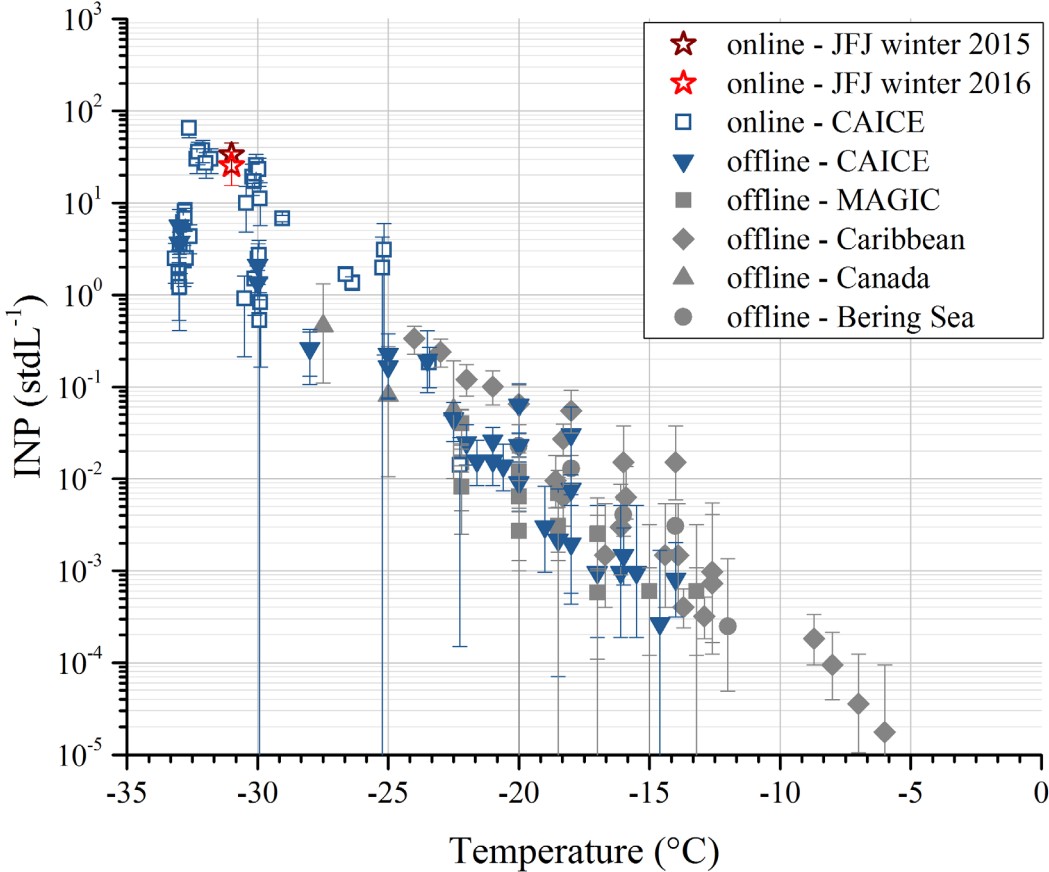

Fig. 12: INP concentrations as function of sampling temperature. Data represent measurements of sea spray particles (DeMott et al., 2016) in the laboratory (blue) at the Center for Aerosol Impacts on Climate and the Environment (CAICE), and for ambient marine boundary layer particles (grey), during different campaigns (see label for respective field campaign name), and two marine events obtained at JFJ (stars, this study). Measurements are differentiated between online (open symbols) and offline (filled symbols) freezing methods. Laboratory data are normalized to total particle concentrations of 150 cm$^{-3}$. Error bars are given for twice the Poisson sampling error, and give up-to-date values, which can differ from published ones in DeMott et al. (2016) (personal communication with the author).





### 4.2.2 Case 6th February 2015: SDE

An increase in INP concentrations was measured on 6th February with values up to 146.2 stdL$^{-1}$ and with a median (mean) concentration of 42.6 (55.4) stdL$^{-1}$ (Fig. 10, panel a). The AF also increased by a factor of 10, up to 1.4x10$^{-3}$ (Fig. 10, panel b). During that time a SDE was detected based on the SSA Ångström exponent criterion (see sect. 3.2). An increase in the

particle concentration 0.4 – 0.8 µm and > 0.8 µm (Fig. 10, panel c) supports the presence of larger mineral dust particles, as well as a decrease in the Na/Al ratio (Fig. 10, panel d) indicating a dusty air mass rich in alumina silicate minerals. Emission sensitivities (Fig. 11, panel b) identify a large area over the central Sahara as a particle source region, and back trajectories calculated for this day (Fig. 11, panel d) show that the air parcel was traveling from the Saharan Desert over the Mediterranean to the Alps. An influence from the ocean on the air mass composition cannot be excluded according to the back trajectories,

as the height of the calculated back trajectories over the Mediterranean Sea was > 950 hPa, which indicates some contact with boundary layer air. In addition, source sensitivities show a possible influence from this region. Chemical analysis of cloud water sampled at JFJ during the arrival of the respective air mass reveals that the f(Sr)$_{ss}$ is low (Fig. 10, panel d), which confirms low marine influence. In addition, the Sr isotopic ratio $^{87}$Sr/$^{86}$Sr is increased to a value of 0.70986, (Fig. 10, panel f), which is an indication for Saharan dust (Capo et al., 1998 and references therein). Furthermore, ratios of Na/Al and Pb/Al were

low, due to an enrichment of Al in the water sample. The eBC mass concentration and the NO$_y$/CO ratio are low (Fig. 10, panel g), which is also support for no anthropogenic influence.

Below water saturation, four measurements of INP concentrations were taken during the dust event, two measurements being below the LOD, and two measurements with increased concentrations of 3 and 8.8 stdL$^{-1}$, as compared to a campaign median (mean) value of $\leq$ 0.2 stdL$^{-1}$ (1.7 stdL$^{-1}$). For conditions below water saturation, dust events yielding higher INP concentrations

at JFJ have been reported before (Chou et al., 2011) and in the Saharan air layer via air craft sampling (DeMott et al., 2003b). At water saturated conditions, this is the first study to clearly show that during a SDE at JFJ an increase in INPs is observed for 242 K, however not a surprising result as an increase in INP concentration was observed during minor influence of Saharan dust (Boose et al., 2016b). We note that at 265 K an increase in immersion INP concentrations at JFJ were not observed during SDEs (Conen et al., 2015) indicating that dust contributes to ice nucleation at colder temperatures as has been previously

suggested in numerous studies (see references in Hoose and Moehler, 2012, Murray et al., 2012, Kanji et al., 2017).

Also an increase in larger particles was observed on 3rd February (Fig. 10, panel c), particularly in the size range 0.1-0.8 µm, however, the SSA increased with wavelength, which is atypical for Saharan dust, and no decrease in the Na/Al ratio was observed. In addition, source sensitivities and back trajectories (see Fig. A2 in the appendix) did not show influence from the Saharan desert. However, during this time construction work on the tunnel systems in the Alps under the JFJ station was

conducted, possibly leading to the abrasion of rocks and the emission of larger particles. These particles were not ice-active, since neither an increase in INP concentration above nor below water saturation was observed.



## 5 Conclusions

This is the first study in which an ice nucleating particle counter of HINC's design has been used to quantify ambient INP concentrations at temperature and RH conditions relevant for mixed-phase cloud formation where both liquid and ice particles can co-exist. We demonstrated that HINC, based on the design of the UT-CFDC (Kanji and Abbatt, 2009), was successfully

deployed to sample ambient INP concentrations. The RH and temperature accuracy was determined for the temperature range 223-263 K and at conditions of sub- and super-saturation with respect to water by observing droplet activation of sulfuric acid particles, and deliquescence of sodium chloride and ammonium sulfate particles. In addition, homogenous freezing of sulfuric acid aerosols at temperatures < 235 K also validated accurate conditions in HINC for ice formation. The uncertainty in INP measurements in HINC arises from the variation in temperature and RH, to which the aerosols in the chamber are exposed to,

and are T ± 0.4 K, and $RH_w$ ± 1.5 % ($RH_i$ ± 3.0 %) for temperatures T > 235 K. For field measurements of INPs with ambient aerosols at 242 K, HINC was characterized for an optimum residence time to maximize the growth time of the ice crystals but avoid particle losses due to gravitational settling in the horizontally oriented chamber. INP concentration measurements with HINC from winters 2015 and 2016 at the JFJ were presented at 242 K for $RH_w$ = 94 % and 104 %. INP concentrations, excluding specific events of high INP concentrations, were on average ≤ 0.2 $stdL^{-1}$ below water saturation, and are within

range of the median INP concentration of 0.1 $stdL^{-1}$ measured at the same site before, with the Portable Ice Nucleation Chamber, PINC (Boose et al., 2016a), during winters 2012-2014. Above water saturation, INP concentrations are in general an order of magnitude higher compared to below water saturation, with a median concentration of 3.7 $stdL^{-1}$ for winters 2015 and 2016 (HINC), and 2.2 $stdL^{-1}$ during winter 2014 (PINC). The small differences in INP concentrations are expected to occur due to natural variability.

In winter 2015, an increase in INP concentrations above water saturation was observed during the influence of an air mass of marine origin, with up to 72.1 INPs $stdL^{-1}$, and a median concentration of 16.3 INPs $stdL^{-1}$. The support of marine influence was based on chemical analysis of cloud water samples with a high $f(Sr)_{ss}$, and an increased Na/Al ratio. Model calculations of back trajectories and air mass origin further support our conclusion of a marine source, but cannot exclude contributions from anthropogenic sources due to entrained particles which can result in chemical ageing processes during transport to JFJ.

Another marine event was identified in winter 2016, when INP concentrations increased to values with up to 176.8. During winter 2014, Boose et al. (2016a) identified a marine influenced air mass arriving at JFJ, but the INP concentrations were within the campaign average, however, with increased ice-active surface site densities, compared to periods when Saharan dust could have been present. Our findings together with those from Boose et al. (2016a) suggest that JFJ could be regularly be affected by marine aerosols, which can therefore contribute to bursts of increased INP populations in the free troposphere.

An air mass with an increase of INPs was sampled during a SDE in winter 2015, and median INP concentrations increased to 42.6 $stdL^{-1}$, with a peak INP concentration of 146.2 $stdL^{-1}$. The identification of the dust laden air mass was supported by several independent measures of aerosol physical and optical properties such as an increase in larger sized particles (0.4 – 0.8 μm and > 0.8 μm), and a negative SSA Ångström exponent, and chemical analysis of cloud water samples, as well as a decrease





in the Na/Al and Pb/Al ratios due to an enrichment in Al. The dust source was further supported by air mass back trajectory and source sensitivity calculations revealing the Saharan desert as a source region.

To extend measurements to warmer temperatures, a significantly improved LOD must be achieved as INP are rarer at warmer temperatures. To quantify INP concentrations in the range between 253 and 273 K, a technique that is sensitive at warmer temperatures must be used, such as offline techniques of drop freezing (e.g. Mason et al., 2015; Conen et al., 2015). However, improving the LOD of online counters can also be achieved by means of using an aerosol concentrator upstream of an INP counter. This has been done in winters 2013 and 2014 (Boose et al., 2016a), achieving a concentration factor of 3. Recently a more efficient aerosol concentrator was implemented during a field campaign at JFJ (winter 2017) and will be the subject of a separate study. The ability to conduct field measurements with HINC will aid future measurements with increasing frequency at JFJ, to determine diurnal, and inter-annual variabilities in INP concentrations at this location.





**Acknowledgements**

This research was funded by the Global Atmospheric Watch, Switzerland (MeteoSwiss GAW-CH+ 2014-2017). We thank the International Foundation High Altitude Research Station Jungfraujoch and Gornergrat (HFJG) for the opportunity to perform the measurements, and the custodians Maria and Urs Otz, Joan and Martin Fischer, Susanne and Felix Seiler for their support

and help. For providing meteorological data we thank MeteoSwiss. Y. Boose and U. Lohmann acknowledge funding from the European Union's Seventh Framework Programme (FP7/2007-797 2013) under grant agreement no. 603445 (BACCHUS). Trace gases measured at JFJ are part of the Swiss National Air Pollution Monitoring Network which is jointly run by EMPA and the Swiss Federal Office for the Environment. This project has also received funding from the European Union's Horizon 2020 research and innovation program under grant agreement No 654109, and was as well supported by the Swiss State

Secretariat for Education, Research and Innovation (SERI) under contract number 15.0159-1. The opinions expressed and arguments employed herein do not necessarily reflect the official views of the Swiss Government. We acknowledge James Atkinson, Robert David, Fabian Mahrt, Nadine Borduas and Claudia Marcolli for useful discussions. For technical support we would like to thank Hannes Wydler, whose expertise greatly helped to improve the instrument.

Author Contributions

LL wrote the manuscript, with contributions from ZAK. ZAK and UL conceived the field study. ZAK and LL designed the laboratory experiments. Field measurements were designed by LL, YB and ZAK. LL conducted all INP measurements and analyzed all INP data. LL, ZAK and UL interpreted the INP data. AZ conducted part of the cloud water sampling, and analyzed and interpreted all the data. EH contributed data on size distributions. NB contributed data on absorption characteristics. MS contributed data on trace gases. ZAK oversaw the overall project.





## Appendix A

### A1 Meteorological conditions

For a complete description of the INP measurements in winter 2015 the meteorological data during the same sampling period are presented in Fig. A1.

Ambient temperatures ($T_{ambient}$) stayed for the whole campaign duration below 0°C, ranging between -9°C to -24°C (Fig. A1, panel b). The sky temperature ($T_{sky}$) is calculated from the longwave radiation measured at the site, and is used to discriminate between in-cloud and out-of-cloud conditions (see Herrmann et al., 2015). During times when the site is in clouds, one would expect the difference between $T_{ambient}$ and $T_{sky}$ to be small, since the longwave radiation received presents the temperature of the cloud surrounding the site and having a similar temperature as the ambient air. INP measurements in- and out-of-cloud conditions do not show significant differences.

The wind velocity (windv) as well as the hourly maximum wind velocity (windvmax) also do not show a correlation with INP concentrations, and a relationship between them is excluded (Fig. A1, panel c), also excluding the role of blowing snow (Lloyd et al., 2015) on our measurements. The wind direction on JFJ (Fig. A1, panel d) represents the two typical wind directions from Northwest and Southeast, which is a result of the orientation of the surrounding terrain. The SDE on 6[th] February was transported in a south-easterly flow, as expected, despite no relation to INP concentration can be concluded. Also the ambient RH does not show an influence on INP concentrations (Fig. A1, panel e).





Fig. A1: Time series of meteorological data, taken by MeteoSwiss: (a) INP concentrations at 242 K and $RH_w$ = 104 % and 94 % as measured with HINC; (b) ambient temperature ($T_{ambient}$) and sky temperature ($T_{sky}$) (Herrmann et al., 2015); (c) hourly maximum wind velocity (windmax) and wind velocity (windv); (d) wind direction; (e) ambient relative humidity ($RH_{ambient}$).





## A2  Back trajectory and source emission sensitivities for 3rd February 2015

During February 3rd 2015 an increase in aerosol particles > 0.5 µm was observed, which was not leading to an increase in INP concentrations below and above water saturation. To exclude influence of Saharan dust particles, which is also indicated in an increase in larger particles, we show here source emission sensitivities (Fig. A2, panel a) and back trajectories (Fig. A2, panel

b) for the respective day, which indicate the air mass arriving at JFJ originated in Norther Europe, and therefore excluding the Sahara as a source region.

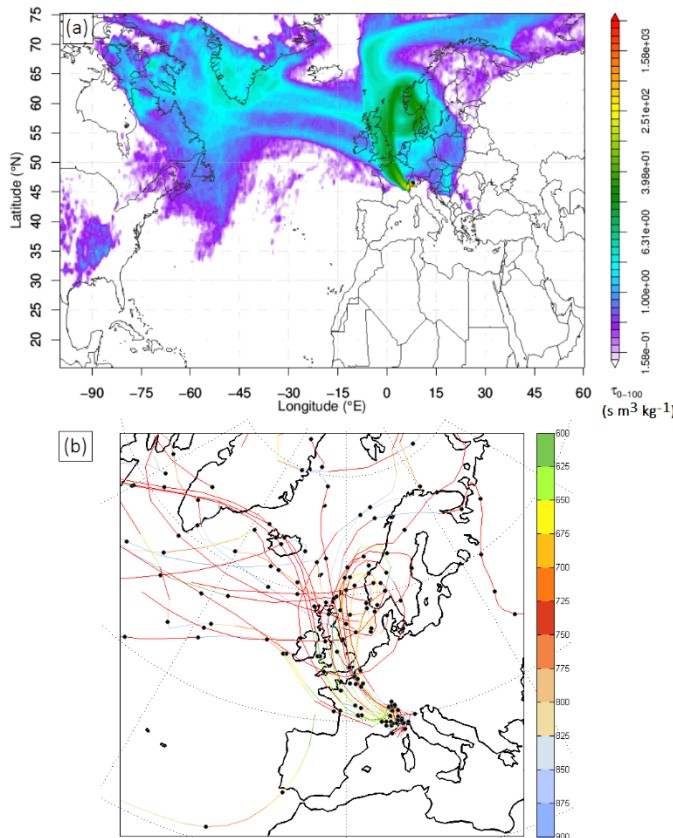

Fig. A2: (a) FLEXPART emission sensitivity fields for 3rd February 2015 calculated 100 m above model ground level (http://lagrange.empa.ch/FLEXPART\_browser/; Stohl et al., 2005; Sturm et al., 2013; Pandey Deolal et al., 2014), colour

code represents the strength of source region contributions to the aerosol burden given in a unit flux per area. (b) 10-day back trajectories calculated with LAGRANTO (Wernli and Davies, 1997); the colour code represents the trajectory pressure above model ground, black points indicate each 24 hour back calculation.



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
