# Peer review of "The Horizontal Ice Nucleation Chamber HINC: INP measurements at Conditions Relevant for Mixed-Phase Clouds at the High Altitude Research Station Jungfraujoch"

_Atmospheric Chemistry and Physics, 2017_

## Referee Comment (RC1) · Anonymous Referee #1 · 3 Jul 2017

**Review of "The Horizontal Ice Nucleation Chamber HINC" (by Larissa Lacher, et al.)**

The paper presents a new ice nucleation counter (HINC) for measurements of ice nucleating particles at conditions relevant for mixed-phase clouds at the Jungfraujoch. Generally the paper is well written with few typos – I have hardly any issues with the approach and style of presentation. However, I have some concerns about what the measurements actually show and how they can be used quantitatively, so I have focussed on these points in my review. If left unaddressed I feel that the paper will potentially lead to a great deal of confusion between various groups working in similar areas.

Firstly figures 2, 3, 4, and 5 present scans from the HINC instrument. The y-axis is labelled AF, which I assume is short for "activated fraction"; however, I do not think this is a good description because they each refer to different things: for example, figure 2 is number of ice crystals nucleated; figures 3 and 4 are fraction of water drops above a certain size (although a small fraction could still be ice crystals); figure 5 is refers to particles that grow above a certain size and eventually activate. Below 100% particles will not be activated in any case, so these should not be referred to as activated fraction.

Figure 2 from the paper presents the homogeneous freezing curve of 100nm dry diameter H2SO4 particles at 233K (-40C), which according to Koop et al. (2000) will freeze when the RH increases and the particles take up water and become dilute enough. I have reproduced the figure from the paper below (Figure 1) and superimposed the fraction of ice crystals nucleated (calculated with a model that uses the equations for homogeneous nucleation rate by Koop et al. 2000).

The model assumes that there is a ramp in RH that occurs over 100 seconds. This time period was purely arbitrary for demonstration purposes, although with a shorter ramp in RH (8 seconds) the onset of nucleation occurs even later and has a lower peak.

In the calculations at 99% RH the physical size of the particles is approximately 0.4 microns diameter, corresponding to a diameter growth factor of 4 and a liquid water volume of approximately 3e-20 m$^3$.

According to the Koop et al. (2000) paper, the nucleation rate under these conditions is approximately 1.51e+16 m$^{-3}$ s$^{-1}$, so we may estimate the fraction of droplets frozen in 1 second as:

$$f = 1 - \exp\left(-1.51 \times 10^{16} \times 3 \times 10^{-20} \times 1\right)$$

which is around 5e-4 for the activated fraction.

The above equation for frozen fraction is derived from the usual 1[st] order rate equation used to approximate homogeneous nucleation (J is the nucleation rate and V is the drop volume):

$$\frac{dN}{dt} = -NJV$$

The full results of this time-dependent calculation are shown in Figure 1, below. The blue line assumes that the Van't Hoff factor is 2, while the green line assumes it is 3 (to span a range relevant to H2SO4). As can be seen there is a marked difference between the theoretical curve and the measurements. The measurements suggest that the ice crystals nucleate starting at around 97% RH, while the theory suggests it is 99% RH. There are also differences in the shape and height of the frozen fraction curve.

I would suggest that these points needs some discussion, otherwise it will lead to confusion in the literature between measurement and models. It is not clear to me whether this is a problem with the theory by Koop et al. or with the measurements presented in the paper.

[Figure]

**Figure 1. Reproduced Figure 2 and theoretical curves superimposed (blue line assumes a van hoff factor of 2, while the green line assumes a Van't Hoff factor of 3).**

Furthermore Figure 3 from the paper presents the "warm activation" curve for 200nm dry size H2SO4 particles. The temperature of 243K (-30C) is high enough so that it is not affected too strongly by homogeneous freezing, so the data represent the growth of aerosol particles into cloud drops. The plot is reproduced below in Figure 2 and shows the activated fraction of 200nm H2SO4 particles given different assumptions about the size of an activated particle.

One can also do a theoretical calculation to infer the critical humidity where particles grow into droplets by finding the maximum value of the Koehler curve (note we have used the assumption of ideality for the activity of water term) .

$$RH_{eq} = \frac{n_w}{n_w + vn_s} \exp\left(\frac{4M_w\sigma}{RT\rho D}\right)$$

for $n$=3; $n_s$=7.86e-17 moles; $s$=0.084 Nm$^{-1}$ (at 243K); $M_w$=0.018 kg mole-1; $R$=8.314 J mole$^{-1}$ K$^{-1}$; $T$=243 K, $r$=1000 kg/m$^3$, a minimisation routine finds that $n_w$=1.11e-11 moles; $D$=7.26e-6 m and $RH_{eq}$=1.00038.

So for dry 200nm (spherical) H2SO4 particles the critical humidity is approximately 100.04%, much less than the >102% suggested by the plot in Figure 2. The above theoretical values are more consistent with text books (see Figure 6.3, page 175, of Pruppacher and Klett, 1997, for dry NaCl for example).

[Figure]

**Figure 2. Reproduced from Figure 3 in the paper, with the green line showing the theoretical critical humidity for the activation of 200nm H2SO4 particles.**

Given the above discrepancies for size selected particles it is difficult to know how to interpret figure 4 from the paper, which is based on passing ambient poly-disperse particles through the instrument.

Finally, figure 5 in the paper shows a humidity scan at 238K for 200 nm NaCl particles. This scan has two traces – one for particles larger than 1 micron and the other for particles larger than 0.3 microns. The "larger than 1 micron" curve approaches an activated fraction for RH > 102 %, whereas the "larger than 0.3 micron" curve approaches an activated fraction for RH > 92% or so. A theoretical calculation (Figure 3) shows that the physical size of 200 nm NaCl particles will always be greater than 0.3 microns for humidities > 80%. If this is the case, the figure 5 should have the activated fraction for 0.3 micron particles equal to 1 for all RH > 80%. Additionally, the physical size of these particles exceeds 1 micron for an RH of 99.5%, not 102% (as suggested by the data in figure 5). Hence, the activated fraction curve should go to 1 for RH = 99.5%, and not 102% (as in the paper).

[Figure]

**Figure 3. Theoretical calculation of the growth of 200 nm NaCl particles. Bottom red line is y=0.3 microns, top red line is y=1 micron.**

I understand that size-selected particles will also have a distribution in size, and this may affect the results slightly. It would be worth discussing the breadth of the size-selected distribution to allow readers to better understand the measurements being presented.

However, at present, given the above difficulties my suggestion is for a major revision of the manuscript to either calibrate any systematic biases, or to give clear reasons for the apparent inconsistencies.

---

## Referee Comment (RC2) · Anonymous Referee #2 · 4 Jul 2017

Larcher et al. describe and characterize a new instrument (HINC) for detecting ice nucleating particles (INPs) in the atmosphere. They then use the instrument to quantify INP concentrations in the deposition mode and immersion mode at a high altitude research station. Concentrations of INPs during two winters are reported and two case studies during high concentrations of INPs are discussed. Since INPs play an important role in climate and the hydrological cycle this topic is well suited for ACP. The paper is well written, and in most cases the results support the conclusions. The

experiments and analyze are also laudable. However, I do have a few major concerns that need to be addressed before I recommend publication.

Major concerns:

Page 12, Line 11-12. The authors suggest that liquid droplets are not detected in the OPC channel > 5 micrometers because they settle out of the aerosol flow and hence are not sampled by the OPC. Is it possible that some of the ice crystals > 5 micrometers also settle out of the aerosol flow and are not sampled by the OPC? If so, does this mean that HINC only measures a lower limit to INP concentrations?

Page 18, Line 8-9: particle loss for 2 micrometer particles is large (44%). What is the transmission efficiency of unactivated particles > 5 micrometers (i.e. 6-10 micrometer particles)? Could a small percentage of unactivated 6-10 micrometer particles be detected as INPs in your experiments and cause experimental artifacts? As an example, would the conclusions in the paper change, if 1% of the 6-10 micrometer particles are unactivated in the HINC and are detected in the OPC channel > 5 micrometers. Could large (6 to 10 micrometers in diameter) primary biological particles cause exprimental artifacts by making it through the HINC unactivated and being detected in the OPC channel > 5 micrometers?

Page 33, line 11-12: Here the authors indicate that HINC avoids particle losses due to gravitational settling in the horizontally oriented chamber. What size of particles are the authors referring to at this point? I find this statement confusing since eariler they indicated that liquid droplets > 5 micrometers settle out and the transmission efficiency of 2 micrometer particles is low.

Page 13, line 4-6: I appreciate that the authors have carried out several systematic studies to determine the upper RH limit for ice crystal detection in the immersion mode. However, I am not completely convinced that at T=242K and RHw < 104 % the OPC size channel > 5 micrometers is well-suited to reliably detect ice crystals in ambient conditions without experimental artifacts. The authors suggest that these experimental conditions are appropriate based on measurements with 200 nm sulfuric acid and 200 nm ammonium sulfate particles, as well as ambient particles. However, perhaps the results from these test cases are not applicable for all air masses encounter at Jungfraujoch. For example, do the results from the test cases apply to 50 nm secondary organic aerosol and 50 nm sea spray particles? What about 800 nm particles? Since the authors have not investigated the effect of particle size or chemical composition (other than sulfuric acid, ammonium sulfate and one ambient situation), I do not know the answer to this question. Also, how representative were the measurements with the ambient particles shown in Figure 4? Were the ambient measurements shown in Figure 5 only carried out on one day or one type of air mass? At T=242K and RHw < 104 % perhaps the OPC size channel > 5 micrometers is not well-suited to reliably detect ice crystals in air masses influenced by marine origin. Additional discussion and possibly additional results are needed to address these questions.

Minor comments:

Abstract, Page 2, Line 17-19: The evidence for marine aerosol acting as INPs is circumstantial. Hence, I think "indicating" should be replaced by "possibly indicating" or "consistent with". In the abstract, the authors should also point out that during the event influenced by marine air, they cannot rule out contributions from anthropogenic or other sources.

Page 4, Line 6-8: This sentence refers to reports more than five decades ago, but then references a paper published in 2014. Either remove the 2014 reference or modify the sentence for consistency.

Figure 2: Indicate in the figure caption that the legend on the right hand side refers to different size channels of the OPC.

Page 20, Line 18: please indicate the purity of the nitric acid solution.

Page 23, Line 29: The authors state that the median concentrations are less than or

equal to 0.2 stdL-1 during winters 2015 and 2016. This statement appears to contradict Fig. 8, where the median values are greater than 0.5 stdL-1 for winter 2015. Am I missing something here?

Page 27, line 19-21: Could primary biological particles also be important for land not covered by snow.

---

## Referee Comment (RC3) · Anonymous Referee #3 · 26 Jul 2017

Authors present a new instrument to measure INP concentration particularly relevant to mixed-phase clouds. They validate the instrument performance using standard chemical compounds for which the thermodynamic properties are well known. They deploy the instrument at the field site to further validate the performance. Results look promising, and I recommend publication after addressing following minor comments. The main concern is how ice crystals are distinguished from droplets, and the discussion over this topic is not convincing.

Can aerosol movable injector perturb the flow conditions within the main chamber?

The injector object is obstructing the sheath flow.

The length of the injector is defined but did not find the diameter of the injector. Do the injector has slots to inject aerosols? What are the dimensions of these slots? Do all these slots deliver aerosol particles evenly and how this is calculated?

The distance between the warm and cold plates is 20 mm. Typically within CFDC, it is around 10 mm. Is the gap is maintained at 20 mm because to increase the particle residence time? Would this affect the supersaturation profile?

Figures 2 and 3: It is not clear why different size acid droplets were used. In figure 2 ice can be detected in the OPC size bin 2 – 8 um. In figure 3, where we don't expect ice particles, the droplets can be seen in three bin channels that range from 0.5 – 5 um. The size channels overlap between figure 2 and 3. This is confusing to understand. Is this means one cannot use size channels only to distinguish between ice and supercooled droplets?

This leads to other question regarding figure 4. How one can determine the INP concentration using this data? On page 13 it says one can use size channel > 5 um, but on page 12 (line 11) it says larger particles (> 5 um) may settle and one cannot see any particles in this channel. Both statements are confusing.

Page 13, lines 4-6: The experiment was conducted at these conditions and results are shown in Figure 4. Here it was shown that there are no particles observed in channel > 5 um. What if all these ambient particles were Organics or pure inorganic salts; one can see the data shown in figure 4. But if there are small dust particles, they might induce ice nucleation and grow to size < 5 um. In this case, droplets and ice particles co-exist. Discussion along these lines is necessary. Limitations imposed on INP concentrations by these conditions (page 13, lines 4-6) must be discussed.

Page 17: Authors use field measurements to validate the performance. However, there is an assumption made (although not mentioned in the manuscript) that aerosol

properties (size, composition, morphology) have remained constant over all the years throughout winter. This may not be true and cannot use such data to validate the performance of the chamber. I suggest rephrasing the discussion to say that INP measurements results were comparable (Fig 9) with previous measurements but using different instruments.

To improve the readability I also suggest moving some of the text from section 2, 3 and 4 to supplementary. For example, section 2.3 can be shortened. Also, section 3.3 and 4.2.

---

## Author Comment (AC1) · 2 Oct 2017

Reviewer comments are reproduced in **bold** and our responses in normal typeface; extracts from the original manuscript are presented in *red italic*, and from the revised manuscript in *blue italic*.

**The paper presents a new ice nucleation counter (HINC) for measurements of ice nucleating particles at conditions relevant for mixed-phase clouds at the Jungfraujoch. Generally the paper is well written with few typos – I have hardly any issues with the approach and style of presentation.**

We thank Reviewer 1 for the comments.

**However, I have some concerns about what the measurements actually show and how they can be used quantitatively, so I have focussed on these points in my review. If left unaddressed I feel that the paper will potentially lead to a great deal of confusion between various groups working in similar areas.**

The reviewer's comments and calculations are appreciated, and we believe that responding to these will help clarify the manuscript. We hope by responding to concerns and making the corresponding changes stated below, especially with regard to the validation experiments, that confusions on interpretation will be completely circumvented.

**Firstly figures 2, 3, 4, and 5 present scans from the HINC instrument. The y-axis is labelled AF, which I assume is short for "activated fraction"; however, I do not think this is a good description because they each refer to different things: for example, figure 2 is number of ice crystals nucleated; figures 3 and 4 are fraction of water drops above a certain size (although a small fraction could still be ice crystals); figure 5 is refers to particles that grow above a certain size and eventually activate. Below 100% particles will not be activated in any case, so these should not be referred to as activated fraction.**

The activated fraction is defined in the revised manuscript on page 8, lines 2 – 3:

 *"The activated fraction (AF) which is the ratio of aerosol particles which activated into cloud droplets or nucleated ice crystals, respectively, to the number of total particles…"*

Thus we believe that it is suitable to describe Fig. 2, showing the fraction of nucleated ice crystals, and Fig. 3 and 4, which refers to the activation of water droplets and possibly nucleated ice crystals. Indeed, the term should not be used for Fig. 5, which shows deliquescence and subsequent growth. The term in the original manuscript on page 8, line 18 *"AF"* has been changed to (revised manuscript, page 9, lines 9 - 10)

*"The observed increase in the particle fraction due to deliquescence and hygroscopic growth compares well to literature results reported to be $RH_w = 77 \pm 2.5$ % (Koop et al., 2000b)."*

as well as in the caption and y-axis labelling of Fig. 5 (revised manuscript page 13, line 2).

**Figure 2 from the paper presents the homogeneous freezing curve of 100nm dry diameter H2SO4 particles at 233K (-40C), which according to Koop et al. (2000) will freeze when the RH increases and the particles take up water and become dilute enough. I have reproduced the figure from the paper below (Figure 1) and superimposed the fraction of ice crystals nucleated (calculated with a model that uses the equations for homogeneous nucleation rate by Koop et al. 2000).**

**The model assumes that there is a ramp in RH that occurs over 100 seconds. This time period was purely arbitrary for demonstration purposes, although with a shorter ramp in RH (8 seconds) the onset of nucleation occurs even later and has a lower peak.**

Here we would like to point out that the particles going through the chamber are in continuous flow, so the particles themselves are not exposed to a ramp but more to a step function between dry conditions prior to entering the chamber (~ RH < 2%), and upon entering the chamber to the respective conditions of RH in the chamber (as indicated on the x-axis of Fig. 2 in the manuscript). The usual time of a relative humidity ramp in HINC is ~20 minutes, but particles continuously flow in and out of HINC thus being exposed to a constant RH for only 8 seconds residence time, i.e. the same particles are not being exposed to a ramp over 8 seconds, but instead are exposed to a constant RH over 8 seconds as such there could be a time limitation for homogeneous freezing.

For clarification, we added to the original manuscript on page 8, line 4 the following sentence (page 8, lines 8 - 11, revised manuscript):

*"In the respective experiments the $RH_w$ conditions in HINC were increased at an approximate rate of 0.5% $min^{-1}$, so that an increase in $RH_w$ of ~10% was achieved over a total time of 20 minutes, which implies that during the 8-second aerosol residence time in HINC, the particles experienced constant RH conditions."*

**In the calculations at 99% RH the physical size of the particles is approximately 0.4 microns diameter, corresponding to a diameter growth factor of 4 and a liquid water volume of approximately 3e-20 $m^3$.**

**According to the Koop et al. (2000) paper, the nucleation rate under these conditions is approximately 1.51e+16 m-3 s-1, so we may estimate the fraction of droplets frozen in 1 second as:**

$f = 1 - \exp(-1.51 \times 10^{16} \times 3 \times 10^{-20} \times 1)$

**which is around 5e$^{-4}$ for the activated fraction.**

**The above equation for frozen fraction is derived from the usual 1st order rate equation used to approximate homogeneous nucleation (J is the nucleation rate and V is the drop volume):**

$dN/dt = -NJV$

**The full results of this time-dependent calculation are shown in Figure 1, below. The blue line assumes that the Van't Hoff factor is 2, while the green line assumes it is 3 to span a range relevant to H2SO4). As can be seen there is a marked difference between the theoretical curve and the measurements.**

**The measurements suggest that the ice crystals nucleate starting at around 97% RH, while the theory suggests it is 99% RH.**

[Figure]

**Figure 1. Reproduced Figure 2 and theoretical curves superimposed (blue line assumes a van hoff factor of 2, while the green line assumes a Van't Hoff factor of 3).**

As shown by the vertical dashed line in Fig. 2, we have indicated the value based on Koop et al. (2000a), for the expected RH for nucleation of 100 nm solution droplets, which suggests the homogeneous freezing threshold for these conditions at RH $\geq$ 98.6%, which is consistent with the RH = 99% suggested by the reviewer. We assumed a fixed nucleation rate of $10^{10}$ cm$^{-3}$s$^{-1}$ (Koop et al., 2000a), which is now added to the revised manuscript on page 8, lines 13 - 14:

*"… based on a fixed nucleation rate coefficient of $10^{10}$ cm$^{-3}$s$^{-1}$ (Koop et al., 2000a), resulting in a RH$_w$ of 98.6%..."*

The RH reported for the chamber on the x-axis of Fig. 2 is a nominal centre RH in the chamber (original manuscript, page 14, lines 11 - 13; revised manuscript page 16, line 12 – page 17, line 1):

*"…there is a temperature variation of $\pm$ 0.4 K across the aerosol layer for the temperature conditions (242 K) used in the field measurements presented here. The variation in temperature causes a variation in RH$_w$ of $\pm$ 1% (RH$_i$ $\pm$ 2%). This translates into a calculated total uncertainty of RH$_w$ $\pm$ 2% (RH$_i$ $\pm$ 3%) at 242 K and RH$_w$ = 104%."*

As such a fraction of the aerosol particles experience a higher RH than the nominal center value reported. In order to visualize this variation in RH including the uncertainty, we plotted the grey area as the total uncertainty at 233 K and RH$_w$ = 98.6% , which is RH$_w$ $\pm$ 2% (RH$_i$ $\pm$ 3.5%) to present both the variation in RH and the uncertainty in RH from temperature in the aerosol layer.

In Fig. 2 of the revised manuscript, the grey area is the associated variation/uncertainty in RH that particles would experience if the chamber were set to the theoretical RH of 98.6% (as indicated by the vertical dashed line). As such it is very likely that the increase in AF seen at 97.5% is indeed due to a fraction of particles within the aerosol layer that are in fact exposed to higher RH of 98-99% required for homogeneous freezing as shown by the vertical dashed line and the lines supplied by the reviewer. We include this discussion now in the revised manuscript on page 8, lines 14 - 20:

*"The reported RH$_w$ on the x-axis in the figure represents the nominal conditions at the center line of the chamber, which is the center of the aerosol layer. Due to the width of the aerosol layer, the particles are exposed to a variation in RH$_w$ ± 1%, and to an uncertainty in RH$_w$ ± 1% due to the temperature uncertainty. The grey shading in Fig. 2 represents this total calculated uncertainty of RH$_w$ ± 2%, for a prescribed RH$_w$ = 98.6%. When the chamber is set to an RH$_w$ = 98.6%, the aerosols can be exposed to a range of 96.6 – 100.6%. Our experiments reveal an increase in the AF of particles between 2 – 8 µm starting at 97.5%, and reaching a plateau value at 99.5%, which is in excellent agreement to the expected range of freezing within the aerosol layer*."

Also the caption of Fig. 2 (revised manuscript on page 10, lines 4 - 5) is updated accordingly:

*"… and the shaded region indicates the calculated range of RH$_w$ and uncertainty to which the particles in the aerosol layer in HINC are exposed to.*"

**There are also differences in the shape and height of the frozen fraction curve.**

Moreover, the RH variation in the aerosol layer also explains the difference in shape of the AF curve i.e., a quasi-step function (i.e. more gentle slope) and progressive AF as a function of RH$_w$ rather than a steep step function of RH$_w$., since the particles are exposed to a distribution of RH, and not a discrete value. Thus some particles activate earlier, at the upper end of the RH$_w$ variationand start growing into a detectable size range of the OPC, whereas a fraction of the particles appear to activate delayed that are exposed to the lower end of the variation in RH$_w$.

There are discrepancies between the observed AF in HINC and the reviewer's modelled AF. The reviewer calculated AFs based on e.g. a physical particle size of e.g. 400 nm at RH$_w$ = 99% and a nucleation rate of 1.5E+10 cm$^{-3}$s$^{-1}$, according to Koop et al. (2000a), and a nucleation time of 1 second.

However, the reported AF in Fig. 2 in the manuscript is measured after a maximum nucleation time of 8 seconds, which is the residence time in HINC. Using 8 seconds, we expect at the given conditions a higher AF than the 5E-4 suggested by the reviewer.

In order to verify this, we performed the same calculations as performed by the reviewer. For the physical size at the respective humidity conditions we use Koehler theory as calculated with the E-AIM (http://www.aim.env.uea.ac.uk/aim/aim.php) and find a physical size of the initial 100 nm $H_2SO_4$ at 99% of 0.33 µm in diameter, which might explain some of the discrepancies compared to the value of 0.4 µm calculated by the reviewer.

The AF results from using the drop diameter of 0.33 µm, homogeneous nucleation rate coefficient of 1.5E+10 cm$^{-3}$s$^{-1}$ and a nucleation time of 1 and 8 seconds are shown in the figure below. We indeed observe differences in the AF between the measured and the modelled AF at 1 second nucleation time, but a good agreement for a nucleation time of 8 seconds. Thus the nucleation time of 1 second is not representative for our measurements in HINC, and explains the observed differences by the reviewer.

[Figure]

Fig. 1.2: AF as a sum as function of RH$_w$ at 233 K for initial 100 nm H$_2$SO$_4$; black markers refer to observed AF in OPC size channel > 0.5 µm in HINC; blue stars are based on a nucleation time of 1 second, and red stars on 8 seconds; grey area refers to the calculated RH variation and uncertainty.

This is now discussed in more detail in the revised manuscript on page 8, lines 20 - 24:

*"As a result of the RH variation, the shape of the AF is not as steep a function of RH$_w$ as the theoretical lines, but rather a steady increase within the range of RH$_w$ to which the aerosol layer is exposed to. According to theoretical calculations (Koop et al., 2000a) at 233 K and 98.6% (99%) RH$_w$ and resulting nucleation rates, we expect for initial 100 nm H$_2$SO$_4$ particles at a residence time of 8 seconds an AF of 0.03 (0.04), which is in agreement with the observed AFs in Fig. 2."*

**I would suggest that these points needs some discussion, otherwise it will lead to confusion in the literature between measurement and models. It is not clear to me whether this is a problem with the theory by Koop et al. or with the measurements presented in the paper.**

Considering the above discussion, the observed differences between theory and experiments presented here are explained by the variation in T and RH conditions in the aerosol layer, and by the nucleation time available for the particles.

**Furthermore Figure 3 from the paper presents the "warm activation" curve for 200nm dry size H2SO4 particles. The temperature of 243K (-30C) is high enough so that it is not affected**

too strongly by homogeneous freezing, so the data represent the growth of aerosol particles into cloud drops. The plot is reproduced below in Figure 2 and shows the activated fraction of 200nm H2SO4 particles given different assumptions about the size of an activated particle.

One can also do a theoretical calculation to infer the critical humidity where particles grow into droplets by finding the maximum value of the Koehler curve (note we have used the assumption of ideality for the activity of water term).

$$RH_{eq} = \frac{n_w}{n_w + vn_s} \exp\left(\frac{4M_w\sigma}{RT\rho D}\right)$$

for $n=3$; $ns=7.86e^{-17}$ moles; $s=0.084$ Nm$^{-1}$ (at 243K); $M_w=0.018$ kg mole$^{-1}$; $R=8.314$ J mole$^{-1}$ K$^{-1}$; $T=243$ K, $r=1000$ kg/m$^3$, a minimisation routine finds that $n_w=1.11e-11$ moles; $D=7.26e-6$ m and $RH_{eq}=1.00038$.

So for dry 200nm (spherical) H2SO4 particles the critical humidity is approximately 100.04%, much less than the >102% suggested by the plot in Figure 2. The above theoretical values are more consistent with text books (see Figure 6.3, page 175, of Pruppacher and Klett, 1997, for dry NaCl for example).

[Figure]

Figure 2. Reproduced from Figure 3 in the paper, with the green line showing the theoretical critical humidity for the activation of 200nm H2SO4 particles.

The experiments with 200 nm H$_2$SO$_4$ particles aims to determine the onset of cloud droplet formation, which should occur slightly above 100% RH$_w$ based on Koehler theory and indicated by the vertical dashed black line in Fig. 3 of the manuscript (Fig. 2 above) and confirmed by the calculations of the reviewer (onset at 100.04% RH$_w$). The RH of 102 %, to which the reviewer refers to, is the RH at which the maximum in the AF is reached, and not the RH at which the particles start to activate into cloud droplets and grow to detectable sizes. Because of the RH variation (and uncertainty) in the chamber, we expect a fraction of the particles to activate later than the nominal RH indicated on the x-axis (see discussion above), but what is crucial is that a change in the size is observed for a fraction of the particles which are indeed exposed to the nominal centre RH (indicated on the x-axis). To clarify this confusion, we modified the manuscript on page 8, line 26 - 27:

*"We thereby refer to the onset of cloud droplet formation, hence the first observed increase in the AF at a given size which represents cloud droplets."*

Also, captions of Fig. 3, 4 and 5 are updated accordingly by adding (revised manuscript, page 11, lines 3 - 4; page 12, lines 3 - 4; page 13, line 3):

*"Vertical dashed line represents expected onset for cloud droplet formation; …"*

In Fig. 3 an increase of particle concentrations in size channels > 0.5 μm occur at $RH_w$ = 99 %, with a steeper increase at RH = 100%. This is expected as a fraction of the particles in the aerosol layer in HINC are exposed to a RH higher than 99% when the nominal RH of the chamber is set to 99% because of the variation in RH that the aerosol layer is exposed to (as discussed above). We now make this clear in the discussion on page 8, lines 29 - 31:

*"The grey shaded area in Fig. 3, 4 and 5 also includes the calculated variation and uncertainty in $RH_w$ for the given temperature of 242 and 243 K, respectively. As discussed above, the exposure of the aerosol particles to this variable RH lead to deviations from the theoretically calculated critical RH."*

Also the captions of Fig. 3, 4 and 5 (revised manuscript, page 11, line 4; page 12, line 4; page 13, lines 4 - 5) were updated accordingly

*"…grey area refers to the calculated variation and uncertainty of RH in the aerosol layer."*

**Given the above discrepancies for size selected particles it is difficult to know how to interpret figure 4 from the paper, which is based on passing ambient poly-disperse particles through the instrument.**

Fig. 4 aims to determine the onset of cloud droplet formation for ambient particles, which should occur slightly above 100% $RH_w$ based on Koehler theory as stated in the manuscript and confirmed by the calculations of the reviewer (onset at 100.04% $RH_w$) and as indicated by the dashed vertical line.

We note that there is a slight increase in particle concentration in the smaller size channels (0.5 – 1 μm) between $RH_w$ 99-101% but a more steep increase starting at a $RH_w$ of 101% for the OPC size channel 0.5 – 1 μm, followed by an increase in the 1 – 2 μm and 2 – 5 μm channel. The increase in particle activated cloud droplet fraction over the range can be explained by the RH variation and uncertainties in the aerosol layer. Furthermore, we expect to have a significant fraction of particles < 100 nm in the ambient air, and also particles with hygroscopicities lower than that of sulphuric acid, which supports the slightly higher observed RH for activation into cloud droplets at this temperature. To make this point clear, the respective discussion is updated in the revised manuscript on page 8, line 31 – page 9, line 6 (original manuscript page 8, lines 9 – 15):

*"We note that an increase in the AF of initial 200 nm $H_2SO_4$ is observed prior to $RH_w$ = 100% in the 0.5 and 1 μm channels (Fig. 3) which is to be expected due to hygroscopic growth of the $H_2SO_4$ particles. Therefore, the increase in size for $RH_w$ < 100% is only observed in the smaller size channels occurring prior to droplet activation at $RH_w$ = 100%, while an increase at $RH_w$ = 100% in the > 2 μm channel is observed due to cloud droplet activation. On the other hand, the ambient particles show droplet activation in the > 0.5 μm, > 1 μm and > 2 μm channels at $RH_w$ = 101.5% (Fig. 4). This is likely due to the lower hygroscopicity of the ambient particles compared to $H_2SO_4$ and due to a larger fraction of the sampled ambient particles being << 100*

*nm, requiring higher RH for the droplets to activate and grow to detectable cloud droplet sizes at this temperature."*

The following sentence on page 8, line 15 (original manuscript)

*"… but could also be compounded by RH uncertainties (see sect. 2.3)."*

is updated in the revised manuscript on page 9, lines 6 – 7:

*"In addition, the experiments could also be influenced by RH uncertainties (see sect. 2.3)."*

Moreover, prompted by this review, we re-visited performed calculations for particle settling, also taking into account the supersaturation profiles and flow speeds at respective conditions in HINC, which reveal that cloud droplets will only grow to > 5 μm if $RH_w \geq 107\%$. Thus the increase at $RH_w$ 104 - 105% observed in Fig. 4 in the size range > 5 μm is believed to arise from a small fraction of ambient particles that act as INP and form ice crystals heterogeneously at these conditions, since water droplets do not grow to this size at RH = 104%. The previous statement on page 12, lines 10 – 12 in the original manuscript

*"An example of an increase in $RH_w$ to > 106 % at 243 K is shown in Fig. 3, where WDS is not observed in the OPC channel > 5 μm. This is likely due to settling of the larger liquid droplets out of the aerosol flow, which grow to sizes too large to be sampled by the OPC due to the hygroscopic nature of $H_2SO_4$."*

is replaced by (revised manuscript, page 14, lines 2 - 9):

*"Based on diffusional growth calculations (Rogers and Yau, 1989) activated cloud droplets of an initial diameter of 200 nm can grow to a size of 4 μm in HINC at 242 K, $RH_w$ = 104% for a residence time of 8 seconds (conditions used for field experiments reported here), giving us confidence that droplets are not detected in the 5 μm channel. Only at an $RH_w$ of 107% cloud droplets grow to > 5 μm, and therefore by conducting our experiments at $RH_w$ = 104%, we only detect ice crystals in the 5 μm OPC channel. As a confirmation, no counts in the size channel > 5 μm were observed for $H_2SO_4$ particles (Fig. 3) even up to an $RH_w$ of 107%, and only with ambient particles an increase in AF for particles > 5 μm at $RH_w$ = 104 - 105% is observed (see Fig. 4), which can be caused by ice crystals forming heterogeneously, since water droplets cannot grow to this size at the respective conditions in HINC."*

**Finally, figure 5 in the paper shows a humidity scan at 238K for 200 nm NaCl particles. This scan has two traces – one for particles larger than 1 micron and the other for particles larger than 0.3 microns. The "larger than 1 micron" curve approaches an activated fraction for RH > 102 %, whereas the "larger than 0.3 micron" curve approaches an activated fraction for RH > 92% or so. A theoretical calculation (Figure 3) shows that the physical size of 200 nm NaCl particles will always be greater than 0.3 microns for humidities > 80%. If this is the case, the figure 5 should have the activated fraction for 0.3 micron particles equal to 1 for all RH > 80%. Additionally, the physical size of these particles exceeds 1 micron for an RH of 99.5%, not 102% (as suggested by the data in figure 5). Hence, the activated fraction curve should go to 1 for RH = 99.5%, and not 102% (as in the paper).**

[Figure]

**Figure 3. Theoretical calculation of the growth of 200 nm NaCl particles. Bottom red line is y=0.3 microns, top red line is y=1 micron.**

The uncertainty in size selection with the DMA is ± 3.5%, resulting in a size distribution of size selected particles in the range of 200 ± 7 nm. However, in reality this should be much larger, because NaCl particles are not perfectly spherical, thus one can expect a pseudo-mono disperse population. In addition, since the particles are charged prior to passing through a DMA, there exists a substantial doubly or triply charged population of particles (8%, 2%, respectively) meaning that 10% of particles can be in the size range 320-440 nm of the population of NaCl sampled.

Because there is a breadth in the size distribution (as already acknowledged by the reviewer) in addition to the multiply charged particles, a fraction of NaCl particles can grow to sizes larger than 1 μm which is demonstrated by the increase already at $RH_w$ = 80% (in the particles >1 μm trace), with a gradual increase in AF for this trace as the progressively smaller particles in the size distribution grow hygroscopically to sizes larger than 1 μm. This is also true for the particles > 0.3 μm trace. The fact that the increase to an AF of 1 is not a step function like the increase observed at $RH_w$ ~80% suggests that the particle properties are different i.e. the breadth in the size distribution (and the variation in RH experienced by the aerosol layer). The $RH_w$ variation in HINC can only account for a small delay ~2%, which would explain why complete activation as the reviewer states is only observed at $RH_w$~ 102%. In addition, the OPC sizing and counting accuracy at sizes as small as 0.3 μm is of a lower accuracy, since the wavelength of the laser (780 nm) is similar to the size of the particles and thus the absolute values in this case should be less significant and it is more crucial to observe a change in the signal.

Since the purpose of this experiment is to show that we can observe growth at $RH_w$= 80%, and this would only be possible due to the deliquescence RH being surpassed at $RH_w$ = 77%. I.e. the fact that we don't see the onset of the growth (and hence preceding phase change) at 90% or already at 70% suggests that the deliquescence of NaCl should have taken place in the expected range of 77% which could only then have allowed for hygroscopic growth and increase in size that was detected by HINC at $RH_w$ = 80%.

The goal is to demonstrate that within uncertainties, we can prescribe the RH in HINC. As shown, complete activation of the particles is only achieved at $RH_w > 100\%$ (and not at $RH_w = 100\%$) which also informs our decision to perform our immersion/condensation freezing experiments for INP at $RH_w = 104\%$, to be above water saturation for the entire aerosol lamina. We do not suggest that HINC should be used to accurately determine deliquescence RH, but rather use the deliquescence concept to infer that HINC is able to achieve with reasonable accuracy the RH we expect in the center by setting the wall temperature.

This discussion is now included in the manuscript on page 9, lines 11 - 28:

*"We observe a first strong increase in the particle fraction > 0.3 and > 1 µm at 80 - 81%, followed by a gradual increase in the particle fraction to unity (within uncertainties) at $RH_w \approx 94\%$ for the > 0.3 µm trace. Deliquescence is a phase change and not a growth process, and a delay as compared to the literature value (dashed line Fig. 5) is expected, since the deliquesced particles need to grow to a size > 0.3 µm to be detected in the OPC. In theory we would expect all particles to grow to sizes larger than 0.3 µm at $RH_w \geq 80\%$ since the deliquescence and growth threshold has been reached. However, we note that due to an uncertainty in sizing of up to 3.5% in the DMA, particles between 193 and 207 nm for a nominal size of 200 nm will be sampled. In reality, we expect an even broader size distribution because dried NaCl particles are aspherical and result in larger sizing errors (Ardon-Dryer et al., 2015). Due to the size selection method with the DMA, a non-negligible fraction of larger particles (10%) between 320 - 440 nm (from double and triple charged particles) will also be sampled by HINC. This breadth in size distribution may explain the initial increase in particle fraction at $RH_w = 80\%$ arising from the multiply charged particles followed by a progressive increase in the particle fraction up to $RH_w = 94\%$ where all the particles grow to sizes > 0.3 µm. The same can be said for the > 1 µm trace. Note that complete activation in this trace occurs at $RH_w > 100\%$, which is expected from the variation in RH in the aerosol layer. Finally, we note that the goal of this experiment is to demonstrate that HINC can achieve prescribed RH conditions with reasonable accuracy by controlling the wall temperature as is seen by the onset in growth at $RH_w = 80\%$ in Fig. 5. Additionally, we acknowledge that the fraction of particles > 0.3 µm reaches a maximum at higher $RH_w$ than theoretically expected, which can also be attributed to the sizing and counting uncertainty of the OPC, which is most pronounced at these small particle size, when the wavelength of the laser (780 nm) is similar to the diameter of detectable particles."*

**I understand that size-selected particles will also have a distribution in size, and this may affect the results slightly. It would be worth discussing the breadth of the size-selected distribution to allow readers to better understand the measurements being presented.**

We agree with the reviewer's comment and by including the above mentioned statement on page 9, lines 11 - 28 we hope to have addressed this point.

**However, at present, given the above difficulties my suggestion is for a major revision of the manuscript to either calibrate any systematic biases, or to give clear reasons for the apparent inconsistencies.**

We have now given clarifications for the parts that were found to be confusing for the reviewer.

References:

Ardon-Dryer, K., Garimella, S., Huang, Y. W., Christopoulos, C., and Cziczo, D. J.: Evaluation of DMA Size Selection of Dry Dispersed Mineral Dust Particles, Aerosol Science and Technology, 49, 828-841, 10.1080/02786826.2015.1077927, 2015.

Koop, T., Luo, B., Tsias, A., and Peter, T.: Water activity as the determinant for homogeneous ice nucleation in aqueous solutions, Nature, 406, 611-614, 10.1038/35020537, 2000a.

Koop, T., Kapilashrami, A., Molina, L. T., and Molina, M. J.: Phase transitions of sea-salt/water mixtures at low temperatures: Implications for ozone chemistry in the polar marine boundary layer, J. Geophys. Res. Atmos., 105, 26393-26402, 10.1029/2000JD900413, 2000b.

Rogers, R. R., and Yau, M. K.: A Short Course in Cloud Physics, Pergamon, 1989.

---

## Author Comment (AC2) · 2 Oct 2017

Reviewer comments are reproduced in **bold** and our responses in normal typeface; extracts from the original manuscript are presented in *red italic*, and from the revised manuscript in *blue italic*.

**Larcher et al. describe and characterize a new instrument (HINC) for detecting ice nucleating particles (INPs) in the atmosphere. They then use the instrument to quantify INP concentrations in the deposition mode and immersion mode at a high altitude research station. Concentrations of INPs during two winters are reported and two case studies during high concentrations of INPs are discussed. Since INPs play an important role in climate and the hydrological cycle this topic is well suited for ACP.**

**The paper is well written, and in most cases the results support the conclusions. The experiments and analyze are also laudable. However, I do have a few major concerns that need to be addressed before I recommend publication.**

We would like to thank the reviewer for their comments, and address the major and minor concerns individually as presented below.

**Major concerns:**

**Page 12, Line 11-12. The authors suggest that liquid droplets are not detected in the OPC channel > 5 micrometers because they settle out of the aerosol flow and hence are not sampled by the OPC. Is it possible that some of the ice crystals > 5 micrometers also settle out of the aerosol flow and are not sampled by the OPC? If so, does this mean that HINC only measures a lower limit to INP concentrations?**

This is a valid question. We have now performed specific calculations for particle growth and settling taking into account the supersaturation profiles and flow speeds of the operation conditions in HINC as well as growth in the chamber as a function of time, which revealed that particles which activate into cloud droplets and grow to sizes of > 5 μm in the chamber are not lost by gravitational settling. Thus the previous statement on page 12, lines 10 – 12 in the original manuscript

[revised manuscript text omitted]

**Page 18, Line 8-9: particle loss for 2 micrometer particles is large (44%). What is the transmission efficiency of unactivated particles > 5 micrometers (i.e. 6-10 micrometer particles)? Could a small percentage of unactivated 6-10 micrometer particles be detected as INPs in your experiments and cause experimental artifacts? As an example, would the conclusions in the paper change, if 1% of the 6-10 micrometer particles are unactivated in the HINC and are detected in the OPC channel > 5 micrometers. Could large (6 to 10 micrometers in diameter) primary biological particles cause experimental artifacts by making it through the HINC unactivated and being detected in the OPC channel > 5 micrometers?**

According to particle loss calculations, due to the length of the tubing and position of the tubing upstream of HINC, 14 -50% of 5 – 10 µm particles respectively are lost before reaching HINC. Furthermore we performed tests with ambient particles to determine the transmission efficiency of particles through HINC taking into account all upstream tubing. We found that for ambient particles > 5 µm, 0% were detected downstream of HINC. In addition, the concentration for particles > 5 µm is naturally very low at JFJ (on the order of 0.05 stdL$^{-1}$), since these larger particles are not part of the free troposphere, and increases usually only occurs during SDEs (see comment below) and injections from boundary layer air. Therefore we do not believe that unactivated large particles will influence the INP number we report from HINC measurements. We now clarify this in the revised manuscript on page 19, line 19 – page 20, line 2:

*"The experiments revealed a particle loss of 26% for 1 µm particles, and 44% for 2 µm particles, and 100% for particles > 5 µm therefore the OPC channel used to detect ice should not be contaminated with large (> 5µm) unactivated ambient particles."*

And on page 20, lines 4 – 7 of the revised manuscript, we update the statement:

*"In addition, calculations with the Particle Loss Calculator (von der Weiden et al., 2009) revealed that 0.8 % of 1 µm particles, and 2.6 % of 2 µm particles, and 14 – 50% of 5 – 10 µm particles should be lost in the inlet and tubing upstream of HINC which we consider to be negligible in light of the low abundance of ambient particles > 5 µm (on the order of 0.05 stdL$^{-1}$)."*

We appreciate the idea to evaluate the contribution of 1% unactivated particles > 5 µm to INP concentrations, and calculated this based on the size distributions during the campaigns reported here. The highest recorded concentration during the campaigns in the size range > 5 µm, were 0.026 stdL$^{-1}$ and 0.285 stdL$^{-1}$ for winter 2015 and 2016, respectively, both during SDEs. These concentrations are low and do not affect INP concentrations, which are on average 2 – 4 orders of magnitude higher. Implicit in this evaluation is that the larger particles will not activate into ice crystals, which is unlikely.

We include this discussion for winter 2015 in revised manuscript now at page 33, lines 26 – 29:

*"A calculation based on the size distribution of ambient particles from the field campaigns at the JFJ reveals that the maximum contribution of 1% aerosol particles > 5 µm remain*

*unactivated in HINC would be 0.026 stdL$^{-1}$ (0.285 stdL$^{-1}$) in winter 2015 (winter 2016), during a time when INP concentrations reached 85.5 stdL$^{-1}$ (154.5 stdL$^{-1}$). Thus a positive bias of larger unactivated particles to INP concentrations should be insignificant."*

**Page 33, line 11–12: Here the authors indicate that HINC avoids particle losses due to gravitational settling in the horizontally oriented chamber. What size of particles are the authors referring to at this point? I find this statement confusing since earlier they indicated that liquid droplets > 5 micrometers settle out and the transmission efficiency of 2 micrometer particles is low.**

We agree, this is confusing. By particle losses here we meant ice crystal losses and we refer to the optimum residence time, which is an interplay between enough time for nucleation growth against too much time such that growing ice crystals are lost due to gravitational settling, as stated in the original manuscript on page 18, lines 3 - 5 (revised manuscript on page 19, lines 14 - 16):

*"The injector position was set to an optimal residence time (8 sec) for the aerosol particles, which takes into account prevention of ice crystal losses due to gravitational settling in the chamber but yet allowing for enough growth time to reach an optical diameter of ≥ 5 µm."*

The transmission efficiency that the reviewer refers to above for 2 µm being low is for aerosol particles larger than 2µm to enter the chamber given all the tubing and driers upstream of HINC and does not refer to the transmission efficiency and size of ice crystals that nucleate within the chamber.

However, as mentioned in the answer above, we performed now specific calculations and come to the conclusion that at the respective sampling conditions used in the field experiments, which the laboratory experiments aim to validate, no particle settling loss occurs for cloud droplets at 242 K and RH$_w$ = 104% and as such this sentence has been removed (see comment above with Rogers and Yau (1989) reference).

**Page 13, line 4-6: I appreciate that the authors have carried out several systematic studies to determine the upper RH limit for ice crystal detection in the immersion mode. However, I am not completely convinced that at T=242K and RHw < 104 % the OPC size channel > 5 micrometers is well-suited to reliably detect ice crystals in ambient conditions without experimental artifacts. The authors suggest that these experimental conditions are appropriate based on measurements with 200 nm sulfuric acid and 200 nm ammonium sulfate particles, as well as ambient particles.**

**However, perhaps the results from these test cases are not applicable for all air masses encounter at Jungfraujoch. For example, do the results from the test cases apply to 50 nm secondary organic aerosol and 50 nm sea spray particles? What about 800 nm particles? Since the authors have not investigated the effect of particle size or chemical composition (other than sulfuric acid, ammonium sulfate and one ambient situation), I do not know the answer to this question. Also, how representative were the measurements with the ambient particles shown in Figure 4? Were the ambient measurements shown in Figure 5 only carried out on one day or one type of air mass? At T=242K and RHw < 104 % perhaps the OPC size channel > 5 micrometers is not well-suited to reliably detect ice crystals in air masses**

**influenced by marine origin. Additional discussion and possibly additional results are needed to address these questions.**

To clarify and support our results, we include now the above mentioned calculations for diffusional growth of cloud droplets (on page 14, lines 2 - 9), indicating that for initially 200 nm dry size particles water droplets grow to a size > 5 µm only at a $RH_w$ > 107%, when they can be misclassified as ice. However, our measurements should not be affected by this conditions since we conduct our experiments at $RH_w$ ≤ 104%. Using sulfuric acid and ammonium sulfate represents the upper limits in hygroscopicity thus we can be confident that water droplets growing from less hygroscopic ambient particles will not contaminate the 5 µm OPC size channel.

Based on the growth calculations, particles with an initial diameter of 200 nm exposed to 242K and $RH_w$ = 104% can grow to 4.038 µm in diameter, and with an initial particle diameter of 50 nm and 800nm to 4.035 µm and 4.08 µm, respectively, showing that these differences in the initial size are not relevant.

The size and chemistry of the aerosol particles are crucial for the cloud droplet activation, with aerosol size being the dominant factor (as found by Dusek et al. (2006)), however our calculations reveal that the initial particle size is of minor importance for the diffusional growth of the cloud droplets to a size of 5 µm in diameter. Hence the water droplets surviving into the OPC ice detection channel at the given sampling condition is not changing by applying a 50 nm, 200 nm or a 800 nm initial dry diameter, since neither of them can grow by diffusion to 5 µm for our sampling conditions. The chemical effect on activation and subsequent diffusional growth of cloud droplets is negligible, because we run the chamber at $RH_w$ = 104% thus for droplet activation a difference between 50 nm secondary organic aerosol and 50 nm sea spray particle should not be of concern because at these high RH both should activate. In addition, even very hygroscopic particles like sulfuric acid should not reach a 5 µm diameter as discussed in the manuscript.

The same insensitivity of diffusional growth on the initial particle size is found for the diffusional growth of ice crystals: At the same conditions of 242 K and $RH_w$ = 104%, ice crystals growing from particles with initial sizes of 200 nm, 50nm and 800 nm reach a final size of 6.8 ± 0.1 µm. Here the chemistry might play an important role as well, but as it was found in a previous study by Petters et al. (2009), investigating the ice nucleation of biomass burning particles, that these less efficient INPs activated within a residence time of 4 - 5 seconds. This gives us thus confidence by applying a residence time of 8 seconds in our ice chamber, that we are able to detect INPs with a range of efficiencies.

For clarification we add to the revised manuscript on page 14, lines 9 – 15:

*"These calculations also reveal that the diffusional growth of the activated cloud droplets, and hence the final size of the cloud droplets of interest, which is 5 µm for the discussed field experiments, are insensitive to the initial dry diameter of the aerosol particles, since the final droplet size at 242K, $RH_w$ = 104% and a residence time of 8 seconds of an initial 50 nm, 200 nm and 800 nm is 4.035, 4.038 and 4.08 µm, respectively. Also effects of the particle chemistry are assumed to be negligible for the droplet activation, since we conduct our experiments at $RH_w$ ≥ 104% where a variety of aerosol chemical compositions should activate into droplets. Thus water droplets contaminating the 5 µm channel should not occur even with varying hygroscopicities and sizes of aerosol populations."*

By referring to the insensitivity of the final size of cloud droplets to its initial dry diameter, we believe that our results for ice crystal particle concentrations are not biased by any sampled aerosol population, since none of them can grow to a size of > 5μm, and cannot be falsely classified as ice.

**Minor concerns:**

**Abstract, Page 2, Line 17-19: The evidence for marine aerosol acting as INPs is circumstantial. Hence, I think "indicating" should be replaced by "possibly indicating" or "consistent with". In the abstract, the authors should also point out that during the event influenced by marine air, they cannot rule out contributions from anthropogenic or other sources.**

Agreed and changed into *"suggesting"* (now in abstract, page 2, line 20).

The contribution from anthropogenic or other sources to the marine air mass event is now mentioned in the abstract (page, 2, line 17):

*"The contribution from anthropogenic or other sources can thereby not be ruled out."*

**Page 4, Line 6-8: This sentence refers to reports more than five decades ago, but then references a paper published in 2014. Either remove the 2014 reference or modify the sentence for consistency.**

We removed "*Cziczo and Froyd, 2014*" as a reference (revised manuscript, page 4, lines 7 – 8).

**Figure 2: Indicate in the figure caption that the legend on the right hand side refers to different size channels of the OPC.**

Included now in the caption of Fig. 2 on page 10, line 2 - 3:

*"…in the size channel 0.5 – 2 μm (dark grey) and 2 – 8 μm (light grey)."*

**Page 20, Line 18: please indicate the purity of the nitric acid solution.**

We added *"(65%)"* to the revised manuscript on page 22, line 3.

**Page 23, Line 29: The authors state that the median concentrations are less than or equal to 0.2 stdL-1 during winters 2015 and 2016. This statement appears to contradict Fig. 8, where the median values are greater than 0.5 stdL-1 for winter 2015. Am I missing something here?**

This value is the median of 2015 and 2016 together, not to the individual field campaigns. It refers to table 2, which is now explicitly mentioned on page 24, line 21:

*"… for winters 2015 and 2016 (Table 2, third row)."*

**Page 27, line 19-21: Could primary biological particles also be important for land not covered by snow.**

Due to the fact that the ambient temperatures at JFJ were below -20°C at the 2$^{nd}$ February, it is unlikely that biological particles contributed to the measured peak INP concentrations, since they would have been activated at warmer temperatures during transport to the JFJ and subsequently fallen out prior to arrival at JFJ (Conen et al., 2015;Stopelli et al., 2015). This is mentioned in the revised manuscript on page 5, lines 14 – 16:

*"…previous INP activation and subsequent fall-out from advected air masses prior to reaching the JFJ, leaving an air mass which is depleted in INPs upon arrival to the JFJ (Conen et al., 2015; Stopelli et al., 2015)."*

---

## Author Comment (AC3) · 2 Oct 2017

Reviewer comments are reproduced in **bold** and our responses in normal typeface; extracts from the original manuscript are presented in *red italic*, and from the revised manuscript in *blue italic*.

**Authors present a new instrument to measure INP concentration particularly relevant to mixed-phase clouds. They validate the instrument performance using standard chemical compounds for which the thermodynamic properties are well known. They deploy the instrument at the field site to further validate the performance. Results look promising, and I recommend publication after addressing following minor comments. The main concern is how ice crystals are distinguished from droplets, and the discussion over this topic is not convincing.**

We would like to thank the reviewer for the comments and address the minor comments individually below.

**Can aerosol movable injector perturb the flow conditions within the main chamber?**

The critical value for transition between laminar and turbulent flow is given by a Reynolds number of 2000. At given chamber dimensions and at 242 K, the Reynolds number for the total flow in the chamber is 66, and the Reynolds number for the disturbance by the aerosol injector is 880. Since both values are below the critical number, no turbulence is expected within the ice chamber. Furthermore, after the perturbation of the flow by the injector, it is expected that the sheath flow, due to its sandwiching role, stabilizes the flow conditions again, and the initial Reynolds number of 66 is re-established. The flow regime has also been discussed in Kanji and Abbatt (2009) at 223 K, and we now include the calculations at 242 K and for the aerosol injector in the revised manuscript on page 6, line 16 - 21.

*"The outer diameter of the injector is 6.35 mm (inner diameter 3.175 mm) which ensures that at a flow rate of 2.8 stdL min$^{-1}$, and at 242 K, the turbulence regime is not encountered by air flowing over the aerosol injector (Reynolds number 880, well below the threshold of 2000 for turbulent flow). Furthermore, given the chamber dimensions, before and after the injector, the Reynolds number is 66, well below the critical number. It is expected that the minor disturbances in the flow by the injector will not result in transitioning from the laminar to turbulent regime."*

**The injector object is obstructing the sheath flow.**

The sheath flow can pass above and below the injector as it is placed directly in the center of the chamber. The laminar flow should be maintained as discussed in the comment above.

**The length of the injector is defined but did not find the diameter of the injector. Do the injector has slots to inject aerosols? What are the dimensions of these slots? Do all these slots deliver aerosol particles evenly and how this is calculated?**

The inner (outer) diameter of the aerosol injector is 3.175 (6.35) mm (now included in the manuscript, see comment above), and aerosols are released through a slit which has a cross sectional area that is slightly smaller than the area of the entrance to the injector, leading to a small overpressure in the injector in front of the aerosol slit, promoting an equal distribution of the particles over the width of the slit. We include this point for a complete description of the instrument now on page 6, line 14 - 16:

*"The cross section area of the slit is smaller than the cross section of the inner diameter of the injector, which creates a small overpressure at the particles exiting through the slit promoting an equal distribution of the aerosols over the width of the slit."*

**The distance between the warm and cold plates is 20 mm. Typically within CFDC, it is around 10 mm. Is the gap is maintained at 20 mm because to increase the particle residence time? Would this affect the supersaturation profile?**

This is correct, the wider distance was chosen in order to have a wider range of residence times when combined with a variety of injector positions.

The exposure of the aerosol layer to the supersaturation profile at the given chamber dimension of 20 mm dependents on the sheath-to- aerosol flow. When maintaining the sheath to aerosol ratio at 12:1, which is a typical setup in the field, this results in an exposure of the aerosol layer to a variation of $RH_w \pm 1\%$ ($RH_i = 2\%$), which is updated in the revised manuscript on page 16, lines 12 – 13 :

*"…there is a temperature variation of ± 0.4 K across the aerosol layer for the temperature conditions (242 K) used in the field measurements presented here. The variation in temperature causes a variation in $RH_w$ of ± 1% ($RH_i \pm 2\%$)."*

**Figures 2 and 3: It is not clear why different size acid droplets were used.**

Different initial sized particles were chosen according to the aimed experiment: For the activation experiments in Fig. 3 at temperatures > 235 K, larger sized acid particles were selected in order to be closer to the mode size of ambient particles. In Figure 2, the goal was to perform homogeneous freezing experiments to validate the RH and T in HINC, as such it was not necessary to be similar to the conditions we use in the field experiments. Thus we chose 100 nm since that was most convenient for the production and size selection of the particles in our aerosol instruments.

**In figure 2 ice can be detected in the OPC size bin 2 – 8 µm. In figure 3, where we don't expect ice particles, the droplets can be seen in three bin channels that range from 0.5 – 5 µm. The size channels overlap between figure 2 and 3. This is confusing to understand. Is this means one cannot use size channels only to distinguish between ice and supercooled droplets?**

Different OPC size channels were chosen according to the aim of the performed experiments: In the homogeneous freezing experiments we do not expect water droplets in any channels above 0.5 µm, thus all channels above 0.5 µm of the OPC can be used to detect ice.

Fig. 3 refers to cloud droplet activation at 243 K, using $H_2SO_4$, where we expect only cloud droplets to be measured and no ice crystals (homogeneous freezing is insignificant for these

conditions) in any size channels. In order to see the relative growth of cloud droplets to certain sizes, the respective size channels are shown individually. This is specifically done to assess which channels *cannot* be used for detecting ice at 242 K and $RH_w$ =104% (if ice were to be present) merely by demonstrating that water droplets could also grow in to these larger size channels. It is from Fig. 3 that we can say water drops will not contaminate the 5 µm channel thus we can use it to confidently detect ice crystals. But it is also from this type of an experiment that we can say at 242 K and $RH_w$ =104% we cannot use the 3 and 4 µm channel to detect ice crystals for ambient aerosol particles (whose pure composition is not known) as these channels may be contaminated with droplets.

We update the statement in the revised manuscript on page 7, line 21 – page 8, line 2:

*"For the deliquescence and cloud droplet activation experiments, 200 nm particles, and for homogeneous freezing experiments 100 nm $H_2SO_4$ were used. Results from these experiments are presented for different size channels which show the growth of the different particles at different RHs to various sizes. E.g. for the homogeneous freezing experiments at 233K, ice particles are observed in size channels > 0.5 µm at $RH_w$ > 97.5%, while in the experiments at 242/243 K and at $RH_w$ 100 – 107% cloud droplets are measured in size channels < 5 µm, but not in the size channel > 5 µm which is thus used to detect ice crystals."*

**This leads to other questions regarding figure 4. How one can determine the INP concentration using this data? On page 13 it says one can use size channel > 5 µm, but on page 12 (line 11) it says larger particles (> 5 µm) may settle and one cannot see any particles in this channel. Both statements are confusing.**

Fig. 4 is not used to determine INP concentrations, but to observe the survival of water droplets in the OPC channels which are used to determine ice crystals. The settling comment was made with regards to water droplets and not ice crystals (due to the difference in densities). But we clarify this in the revised manuscript.

Still, the reviewer addressed a valid point of inconsistencies regarding the particle settling losses previously mentioned in the manuscript. We have now performed specific calculations of water droplet growth in HINC for the conditions reported here (242 K; up to $RH_w$ 107%) and we find that 200 nm SA particles grow to 5 µm only at $RH_w$ > 107%. Furthermore there is no settling of cloud droplets expected at 242 K and 104% $RH_w$, based on our diffusional growth calculations.

However, since we do see a small signal at 105% in the 5 µm channel for the ambient case (Fig. 4), we believe that this is strong evidence for ice crystals in the ambient case because only ice crystals are expected to grow to sizes larger than 5 µm under these conditions, since water droplets (as shown by the lab experiments with extremely hygroscopic particles) do not reach 5 µm in size below $RH_w$ = 107%. As such we are confident for a RH of 104% we do not detect water drops but only ice crystals in the size channel > 5 µm.

Thus the previous statement on page 12, line 10 – 12 in the original manuscript

*"An example of an increase in $RH_w$ to > 106 % at 243 K is shown in Fig. 3, where WDS is not observed in the OPC channel > 5 µm. This is likely due to settling of the larger liquid droplets out of the aerosol flow, which grow to sizes too large to be sampled by the OPC due to the hygroscopic nature of $H_2SO_4$."*

is replaced by (revised manuscript, page 14, line 2 - 9):

*"Based on diffusional growth calculations (Rogers and Yau, 1989) activated cloud droplets of an initial diameter of 200 nm can grow to a size of 4 µm in HINC at 242 K, $RH_w$ = 104% for a residence time of 8 seconds (conditions used for field experiments reported here), giving us confidence that droplets are not detected in the 5 µm channel. Only at an $RH_w$ of 107% cloud droplets grow to > 5 µm, and therefore by conducting our experiments at $RH_w$ = 104%, we only detect ice crystals in the 5 µm OPC channel. As a confirmation, no counts in the size channel > 5 µm were observed for $H_2SO_4$ particles (Fig. 3) even up to an $RH_w$ of 107%, and only with ambient particles an increase in AF for particles > 5 µm at $RH_w$ = 104 - 105% is observed (see Fig. 4), which can be caused by ice crystals forming heterogeneously, since water droplets cannot grow to this size at the respective conditions in HINC."*

**Page 13, lines 4-6: The experiment was conducted at these conditions and results are shown in Figure 4. Here it was shown that there are no particles observed in channel > 5 µm. What if all these ambient particles were Organics or pure inorganic salts; one can see the data shown in figure 4. But if there are small dust particles, they might induce ice nucleation and grow to size < 5 µm. In this case, droplets and ice particles co-exist. Discussion along these lines is necessary. Limitations imposed on INP concentrations by these conditions (page 13, lines 4-6) must be discussed.**

If there are dust particles that form ice, they will activate upon entering the chamber and grow within the residence time to sizes larger than 5 µm. After nucleation, at the conditions used in the field, particles will grow to 5 µm rather quickly (2 s), compared to cloud droplet growth, because of the difference in $RH_i$ and $RH_w$ (142% and 104%, respectively). As such we should have a negligible contribution from undercounting ice crystals due to particles that nucleate ice but don't grow to 5 µm.

This is now included in the manuscript on page 14, lines 26 - 29:

*"… the residence time of 8 seconds should also minimize the number of ice crystals < 5 µm, since ice crystals only need 2 seconds to grow by diffusion to sizes > 5 µm at this high $RH_i$ = 140%. We believe undercounting INPs due to ice crystals < 5 µm should not significantly influence the INP concentrations reported especially given the day to day variability in INP concentrations found at the field site studied in this work."*

**Page 17: Authors use field measurements to validate the performance. However, there is an assumption made (although not mentioned in the manuscript) that aerosol properties (size, composition, morphology) have remained constant over all the years throughout winter. This may not be true and cannot use such data to validate the performance of the chamber. I suggest rephrasing the discussion to say that INP measurements results were comparable (Fig 9) with previous measurements but using different instruments.**

We agree with the reviewer, and rephrased the revised manuscript on page 24, line 23 – 24:

*"INP concentrations were comparable in different years, given that the minimum and maximum INP concentrations below water saturation overlap."*

**To improve the readability I also suggest moving some of the text from section 2, 3 and 4 to supplementary. For example, section 2.3 can be shortened. Also, section 3.3 and 4.2.**

We believe that the mentioned sections are required to understand the presented results in a complete format, and necessary to understand how the chamber works, as such we have opted to retain it in the main section of the manuscript.

References:

Kanji, Z. A., and Abbatt, J. P. D.: The University of Toronto Continuous Flow Diffusion Chamber (UT-CFDC): A Simple Design for Ice Nucleation Studies, Aerosol Sci. Tech., 43, 730 - 738, 10.1080/02786820902889861, 2009.

Rogers, R. R., and Yau, M. K.: A Short Course in Cloud Physics, Pergamon, 1989.

---

## Author Response (AR2)

Reply to reviewer #1: Suggestions for revision or reasons for rejection of "The Horizontal Ice Nucleation Chamber HINC: INP measurements at Conditions Relevant for Mixed-Phase Clouds at the High Altitude Research Station Jungfraujoch" by Larissa Lacher et al."

Reviewer comments are reproduced in **bold** and our responses in normal typeface; extracts from the revised manuscript in *blue italic*.

**I have read through the reviewer's reply to my suggestions / comments.**

**It was useful to have the explanation that we need to consider the particles experiencing "constant" conditions for approximately 8 seconds, that each particle does not experience the "mean RH", and also the explanation**

**about deliquesce occurring but not appearing in the measurement (because of the same problem). These difficulties are acknowledged by the authors and explanations are given.**

**However, I still feel the authors need to be careful that some of the figures are not over interpreted by readers, and caveat them accordingly, given that the errors in RH are such that it is difficult to quantitatively assess homogeneous nucleation and droplet activation when comparing with theory. Instrument scientists working**

**in this area may be well aware of these difficulties, but modellers may not.**

**Given the discussion about particles not experiencing the "mean humidity" the implications therefore are that the instrument would generally underestimate both the amount of droplet activation and homogeneous nucleation at a measured RH; however, I don't think this is actually said in the manuscript - not clearly anyway - for instance perhaps it should be brought more to the fore in the conclusions. This is the main point I have.**

We dedicate an entire section of the manuscript to the uncertainties and variation of RH in the chamber (see section 2.2.2.) where we discuss the range of RH experienced by the particles. In addition, each figure where AF curves as a function of RH are shown, we explicitly show a representation of this uncertainty by grey or hashed shading. We believe this is a very clear indication to the reader. Also, it is not true that the instrument always underestimates the amount of nucleation/activation because the droplets could be exposed to e.g. $RH_w \pm 2\%$

because of the range of $RH_w$ experienced. See figure below for example where theoretically calculated AF using nucleation rates from Koop et al. (2000) are lower than those measured in HINC. The AF that theoretical calculations would predict at $RH_w$ = 99.9, we already measure those at $RH_w$ = 99%. To clearly emphasize this we added the following in section 2.2.4, page 15, lines 13 – 16:

*"As can be seen from Fig. 6, some deviation (all within instrument uncertainty) of the data points does occur. This is due to the fact that in the region of ice nucleation or droplet activation the AF (nucleation rate) is a very steep function of RH and a small change in water activity by for example 0.01 (~RH of 1% at equilibrium) can change the nucleation rate by a factor of 10 or 15."*

We do agree with the reviewer, that this can be mentioned more clearly in the conclusions and we now do this on page 34, lines 13 – 18:

*"The uncertainty in INP measurements in HINC arises from the variation in temperature and RH, to which the aerosols in the chamber are exposed to, and are T ± 0.4 K, and $RH_w$ ± 1.5% ($RH_i$ ± 3%) for temperatures T > 235 K. These variations can lead to uncertainties in activated fractions in the region of activation and nucleation where nucleation rates are a steep function of RH. Conducting field measurements of INPs with ambient aerosols at 242 K and RH 104%, where we ensure all particles are fully activated to droplets, reduces the associated uncertainties in particle concentrations arising from RH variation in the chamber."*

**My specific replies to previous points raised are that:**

**1. In response to my comment about figure 2: "Our experiments reveal ... excellent agreement". I would not call this "excellent" agreement. By those arguments, given the range in uncertainty in RH you could have presented any value between zero and 1 for the activated fraction and called it excellent agreement. I would say that the general behaviour is consistent with that expected, but it is difficult to use it quantitatively due to errors.**

We agree with the reviewer and deleted "excellent" on page 8, line 19.

**2. Figure 1.2 in the reply. I also do not agree with the red and blue stars:**

**Koop et al (2000) suggests the nucleation rate is around 1.5e16 m-3 s-1 at these conditions. The volume of water on the particles is around 1.8e-20 m^3 so this suggests a frozen fraction of 2e-3 for a residence time of 8s (an order of magnitude less than in their Figure).**

**Please note that my calculations also had the same range of growth factors predicted EAIM (using 2 for the van Hoff factor gives a growth factor of 3, using 3 gives a growth factor of 4, so there is no discrepancy in that calculation)**

We agree with the reviewer and found an error in our code. The nucleation AF and water activities were interchanged in our previous calculations arriving at the wrong AF using theoretical calculations. We thank the reviewer for their thorough comments and have now presented below the theoretical AFs of 1 and 8 seconds. We agree with the reviewer that the calculated AF at 233 K and $RH_w$ = 98.6% is one order of magnitude lower compared to what we observe in HINC for 8 seconds (and 2 orders of magnitude lower for 1 second). As noted by the given uncertainties of $RH_w$ ± 2%, this would represent a shift in RH of 1 % (water activity of 0.1). Because nucleation rate is such a steep function of water activity in this region, a small uncertainty or range of RH experienced results in this change in AF. We now acknowledge this on page 15, lines 13 – 16 (section 2.2.4) and page 34, line 13 – 18 (in the conclusions).

[Figure]

Figure 1: AF as a sum as function of $RH_w$ at 233 K for initial 100 nm $H_2SO_4$; black markers refer to observed AF in OPC size channel > 0.5 µm in HINC; blue stars are based on a nucleation time of 1 second, and red stars on 8 seconds; grey area refers to the calculated RH variation and uncertainty; droplet volumes are calculated with E-AIM Koehler model (http://www.aim.env.uea.ac.uk/aim/aim.php), and nucleation rates are inferred from Koop et al. (2000a).

**3. The liquid activation curves (figure 3 in the manuscript), have the potential to be confusing. They show particles activating up to an RH of 102%. However, from another point of view they show how the instrument behaves and hence are useful if presented in the right way, with a discussion of time-scales.**

In order to clarify the reviewer's concern we have now added the following on page 8, line 30 – page 9, line 3:

[revised manuscript text omitted]